# Covariance patterns between sleep health domains and distributed intrinsic functional connectivity

Yulin Wang [1,2], Sarah Genon [3,4], Debo Dong [2,3], Feng Zhou[2], Chenyu Li[5], Dahua Yu[6], Kai Yuan[7], Qinghua He [2], Jiang Qiu[2], Tingyong Feng [2], Hong Chen[2] & Xu Lei [1,2] ✉

Sleep health is both conceptually and operationally a composite concept containing multiple domains of sleep. In line with this, high dependence and interaction across different domains of sleep health encourage a transition in sleep health research from categorical to dimensional approaches that integrate neuroscience and sleep health. Here, we seek to identify the covariance patterns between multiple sleep health domains and distributed intrinsic functional connectivity by applying a multivariate approach (partial least squares). This multivariate analysis reveals a composite sleep health dimension co-varying with connectivity patterns involving the attentional and thalamic networks and which appear relevant at the neuromolecular level. These findings are further replicated and generalized to several unseen independent datasets. Critically, the identified sleep-health related connectome shows diagnostic potential for insomnia disorder. These results together delineate a potential brain connectome biomarker for sleep health with high potential for clinical translation.

It is increasingly recognized that sleep health (SH) is a multi-dimensional construct[1,2]. Sleep health has been defined as "a multi-dimensional pattern of sleep-wakefulness, adapted to individual, social, and environmental demands, that promotes physical and mental well-being"[3]. Resting on this theoretical consideration and previous researches examining associations between different sleep measures and health outcomes[3–5], sleep health mainly consists of six domains of sleep and circadian functioning: Regularity in sleep, Satisfaction with sleep/sleep quality, Alertness during waking hours, Timing of sleep, Sleep Efficiency/Continuity, and Sleep Duration (Ru-SATED). Emerging studies also pointed out that there exist other domains that can be considered as relevant to the construction of

sleep health, such as sleep quality[6], insomnia symptoms[7], and sleep medication use[7].

Recent studies have begun to construct a sleep health composite towards studying multiple domains of sleep[1,8–10]. Indeed, previous research has mostly focused on single indicators of sleep health, making it difficult to provide a consistent guideline for research and practical settings[6]. A composite measure not only would provide a more comprehensive indicator of sleep health than any single standard measure[11–13] but also would be less prone to noise and hence could present a greater neurobiological validity[14]. For instance, Dalmases and colleagues reported that the Ru-SATED score was more strongly linked with self-rated health status than sleep duration

[1]Sleep and NeuroImaging Center, Faculty of Psychology, Southwest University, Chongqing, China. [2]Key Laboratory of Cognition and Personality, Ministry of Education, Faculty of Psychology, Southwest University, Chongqing, China. [3]Institute of Neuroscience and Medicine, Brain & Behaviour (INM-7), Research Centre Jülich, Jülich, Germany. [4]Institute for Systems Neuroscience, Heinrich Heine University Düsseldorf, Düsseldorf, Germany. [5]Sleep Center, Department of Brain Disease, Chongqing Traditional Chinese Medicine Hospital, Chongqing, China. [6]Information Processing Laboratory, School of Information Engineering, Inner Mongolia University of Science and Technology, Baotou, Inner Mongolia, China. [7]School of Life Science and Technology, Xidian University, Xi'an, Shanxi, China. ✉e-mail: xlei@swu.edu.cn

alone[15]. Nonetheless, considerations of constructing a sleep health composite with multi-domains of sleep health rather than individual sleep metrics is still an emerging research area with many remaining gaps in the literature[14]. In particular, the neurobiological validity of a composite score could be importantly increased by capitalizing on its association with intrinsic functional connectivity in the population.

Resting-state networks are typically derived from the connectivity profile of spontaneous fluctuations in functional MRI (fMRI) signals and are thought to reflect the intrinsic brain functional connectivity[16,17]. Converging research evidence across datasets, methods, and laboratories has agreed upon the principle resting-state networks such as the somatomotor, visual, default mode, and fronto-parietal control networks[18,19]. In that context, resting-state functional connectivity (RSFC), such as the anti-correlations between the default mode network (DMN) and the task-positive network (TPN) was found to be a reliable biomarker to classify different sleep stages [for a review, see[20]]. Also, RSFC was found to be influenced by sleep deprivation evidenced by the reduced connectivity within the DMN[21], the dorsal attention network (DAN), and the auditory, visual, and motor networks [for a tabular overview, see ref.[16]]. Moreover, abnormalities in functional network modules subserving hyperarousal, salience, sensory-motor, cognitive, and self-referential processes, including the limbic, thalamus, sensory-motor, fronto-parietal control, and default mode networks have been shown in insomnia disorder [for a review, see refs.[22,23]]. These lines of research together suggest RSFC contains crucial information underlying the neurobiological mechanisms of different aspects of sleep health.

Actually, RSFC has been shown to be associated with some domains of sleep health such as sleep duration[24], sleep quality[25] and timing of sleep (or chronotype)[26]. Despite this increasing interest in revealing the interplay between intrinsic brain functional connectivity and domains of sleep health, existing studies have been limited in

several respects. First, most have adopted a categorical approach, or only examined a single domain of sleep health, and are therefore unable to capture the heterogeneity across different sleep health domains. Second, constructions of sleep health composite were mainly driven by the conceptual Ru-SATED model, rather than being guided by the intrinsic structure of the brain and behavior features. Third, existing work in sleep health composite has often used relatively small samples (e.g., dozens of participants)[27]. While multivariate techniques allow the examination of both multiple brain systems and sleep health domains simultaneously, such techniques usually require large samples[28,29]. Also in line with these considerations, the high dependence and interaction across different domains of sleep health (e.g., short sleep duration is usually accompanied by lower efficiency and regularity, lower sleep satisfaction/quality) encourages a transition in sleep health research[14] from categorical to dimensional approaches that integrate intrinsic brain functional connectivity and sleep health.

In this study, we addressed this research gap by relating RSFC to a large set of sleep health measures in a single integrated analysis to identify a brain connectome biomarker for sleep health. More specifically, we addressed the following four research questions (Fig. 1): (1) Can we identify a composite sleep health dimension that relates to RSFC patterns? (2) Can the obtained dimension be replicated and generalized to unseen independent datasets? (3) A growing number of studies highlights the relationships between alterations of some neurotransmitter systems including the serotonin receptors, glutamate, and γ-aminobutyric acid (GABA) and sleep disturbances or "'unhealthy" sleep[30–32]. Accordingly, we here examined whether the brain connectome underlying interindividual differences in sleep phenotype may be related to specific neurotransmitter systems (based on neurotransmitter receptor maps[33]). (4) Does the sleep-health-related brain connectome have predictive utility for both the health population and insomnia disorder?

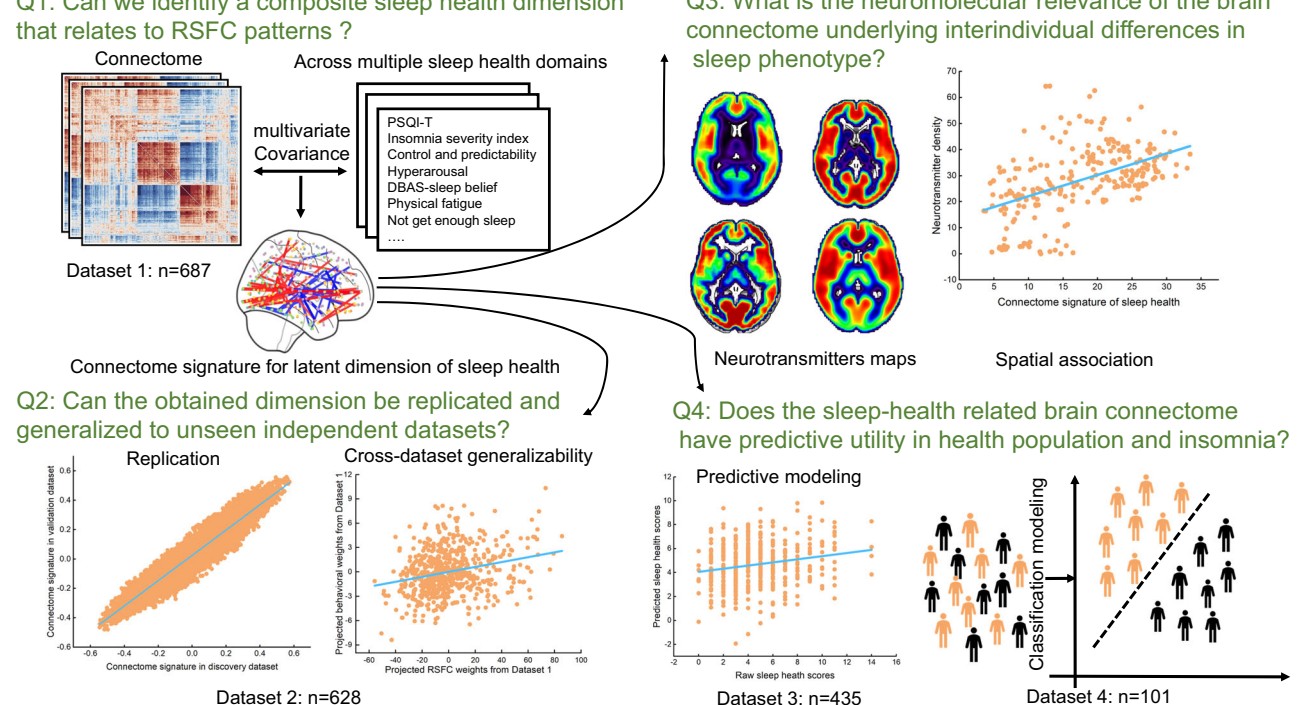

**Fig. 1 | Overview of research questions and main analyses.** This study aims to answer four research questions (Q1–4) to identify a brain connectome biomarker for sleep health. To do so, this study used the multivariate and machine-learning approach (i.e., partial least square, predictive modeling, classification) with four independent datasets.

To answer these questions, we first have capitalized on a large sample of young adults from the Behavioral Brain Research Project to Chinese Personality (BBP) by applying a machine-learning technique called partial least squares (PLS)[34–36]. As a multivariate data-driven statistical technique, PLS is capable of discovering the latent dimensions that link interindividual variability in intrinsic functional organization to interindividual variability in behavioral measures spanning multiple domains of sleep health. To comprehensively assess this latter, we included measures suggested by the Ru-SATED model, but we also incorporated several additional metrics associated with sleep health defined broadly[6], that are: (1) beliefs, attitudes, and habits about sleep, considering that individuals with more-positive beliefs and attitudes about sleep or who adopt sleep-protective behaviors are more likely to experience better sleep health[37]; (2) sleep deficiency which also incorporates insomnia symptoms, sleep disorders and sleep medication use, considering both sleep health and sleep deficiency can be regarded as the anchors at either end of a continuum[3]; (3) have one slept enough, whether one felt refreshed upon waking[38], the necessity of nap and the needed nap time were also incorporated to evaluate the satisfaction with sleep/sleep quality[6]; (4) tendency to engage in spontaneous waking thought[39], attention-related deficiency[40] and fatigue[41] as supplements to assess the alertness during waking hours. To ensure that the latent dimensions linking sleep phenotype to RSFC uncovered by PLS were robust, multiple control analyses were performed. Furthermore, as multivariate approaches in high-dimensional data such as PLS are prone to overfitting, a 10-fold cross-validation was performed here to assess the generalization performance of the latent dimensions to unseen test data.

As described below, we uncovered one latent dimension of the RSFC that was highly correlated with one latent dimension of sleep health in a discovery dataset ($n = 687$). This dimension was characterized by a specific spectrum of sleep health-related measures and by specific RSFC features. This latter appeared as a neurobiologically relevant connectome since the RSFC loadings were spatially correlated with the distribution of several neurotransmitter systems relevant to sleep health, including serotonin receptors, as well as metabotropic glutamate receptor 5 (mGluR5) and the γ-Aminobutyric acid type A (GABAA) receptor[30,31]. Furthermore, the associative pattern was successfully replicated, and the model showed generalizability in an internal validation dataset ($n = 628$). Importantly, the identified connectome pattern shows predictive utility for sleep quality in an independent sample consisting of unrelated individuals from the Human Connectome Project (HCP) dataset ($n = 435$). Finally, this connectome pattern critically shows diagnostic potential to distinguish insomnia patients ($n = 52$) from sleep-healthy subjects ($n = 49$) with an accuracy of 79.12%. These results hence delineated a sleep health dimension whose associated functional connectome could serve as an objective neuroimaging biomarker in clinical translations, such as to assess sleep-based interventions for improving brain health.

## Results

### PLS model reveals one robust dimension linking sleep health and resting-state function connectivity

We sought to delineate multivariate relationships between resting-state functional connectivity and sleep health in a large discovery dataset ($n = 687$). To this end, we applied PLS, an unsupervised machine-learning technique that seeks to find covariance between two high-dimensional matrices, namely whole-brain RSFC and 36 behavioral measures spanning multiple domains of sleep health. Following preprocessing using a validated pipeline that minimizes the impact of in-scanner motion (see "Methods" section), we constructed subject-level RSFC using a 246-node parcellation system[42] that includes 210 cortical regions and 36 subcortical regions. Prior to analysis with PLS, we regressed age, sex, handedness, and head motion out of both the RSFC and behavior data to ensure that these potential confounders did not drive results. The input data thus consisted of 30,135 unique functional connections and 36 sleep health-related variables. The selection of the sleep health measures was based on the consideration of selecting available variables in the BBP that (1) represent central domains of SH described in the Ru SATED[3] as well as in the National Sleep Foundation (NSF)'s Sleep Health Index[6]; (2) provide high consistency with previous SH studies[5,7,11]; (3) draw a multifaceted picture of SH to a bigger extend based on its broader definition. On that basis, the chosen 36 variables were grouped into seven domains including (1) Satisfaction with sleep/ Sleep quality; (2) Alertness during waking hours; (3) Timing of sleep; (4) Sleep efficiency/ continuity;5) Sleep duration; (6) Sleep deficiency; (7) Sleep beliefs, attitudes, and habits (see "Methods" and Table S1 for the details).

Figure 2a shows the amount of covariance explained by each latent variable (LV). Notably, only one LV (LV1) survived after permutation testing with FDR correction ($q < 0.05$) (Fig. 2b). Importantly, several control analyses were performed to ensure the robustness of the obtained LV1. See Table S2 for details. First, 10-fold cross-validation was successful; PLS components estimated from 90% of the participants successfully generalized to the remaining 10% of participants, as indicated by the significant correlation between RSFC and behavioral composite scores in the test folds (LV1, mean $r = 0.17$, permuted $p < 3.0 \times 10^{-3}$). It should be noted that to avoid data leakage issues, the adjustment for confounds and data standardization were performed within the cross-validation loop (i.e., at first, we estimated parameters of data standardization and confounds regression in the nine training folds and then applied the obtained parameters to the test fold). Second, PLS components were robust to global signal regression, total intracranial volume regression, time (hour) of acquisition regression, the pre-scanning positive and negative affect regression, body mass index (BMI) regression, and family income regression, as indicated by the high correlation ($r > 0.93$, Table S2) between saliences of original PLS and PLS with corresponding variable regression. Third, instead of regressing age, sex, handedness, and head motion from the data, these variables were added to the phenotypic data for the PLS analysis. The results were largely unchanged as indicated by the high correlation ($r > 0.92$, Table S2) between the saliences of original PLS and PLS with adding age, sex, handedness, and motion into the phenotypic data. Fourth, PLS components were not driven by the non-Gaussian distributions of the behavioral data and skewed behavioral distributions, as indicated by the high correlation ($r > 0.98$, Table S2) between saliences of original PLS and PLS with quantile normalization to improve the Gaussian distributions of the behavioral data. Fifth, the results remained largely unchanged when using a different Seitzman et al.' Atlas[43] containing 300 regions for the RSFC construction, as indicated by the high correlation ($r = 0.99$ for behavior data; $r = 0.78$ for RSFC data) between loading scores of Brainconnectome Atlas and Seitzman et al.' Atlas, see details in Supplementary results, Figs. S1 and S2. Sixth, to demonstrate the sleep health component was independent of both the circadian timing of acquisitions and the state of alertness of individuals during the resting-state fMRI acquisition, we extracted the dimension score of "Sleepiness" from the Amsterdam Resting-State questionnaire (ARSQ)2.0[44] and time (hour) of acquisition, then correlated them with both the RSFC and behavior composite scores for the discovery dataset. The results revealed no significant correlation between participants' sleepiness and either their RSFC composite score ($r = -0.0349$, $p = 0.3605$) or their behavior composite score ($r = -0.0223$, $p = 0.5603$). Additionally, no significant relationship between the acquisition time and either the RSFC composite score ($r = 0.0395$, $p = 0.3013$) or the behavior composite score ($r = -0.0393$, $p = 0.3941$) was observed. Seventh, the first latent component was robust across analytical approaches, which was evidenced by the high correlation between the first principal component of the sleep health behavioral measures (obtained by principal component analysis) and the behavioral saliences of LV1 ($r = 0.91$, $p = 1.22 \times 10^{-14}$).

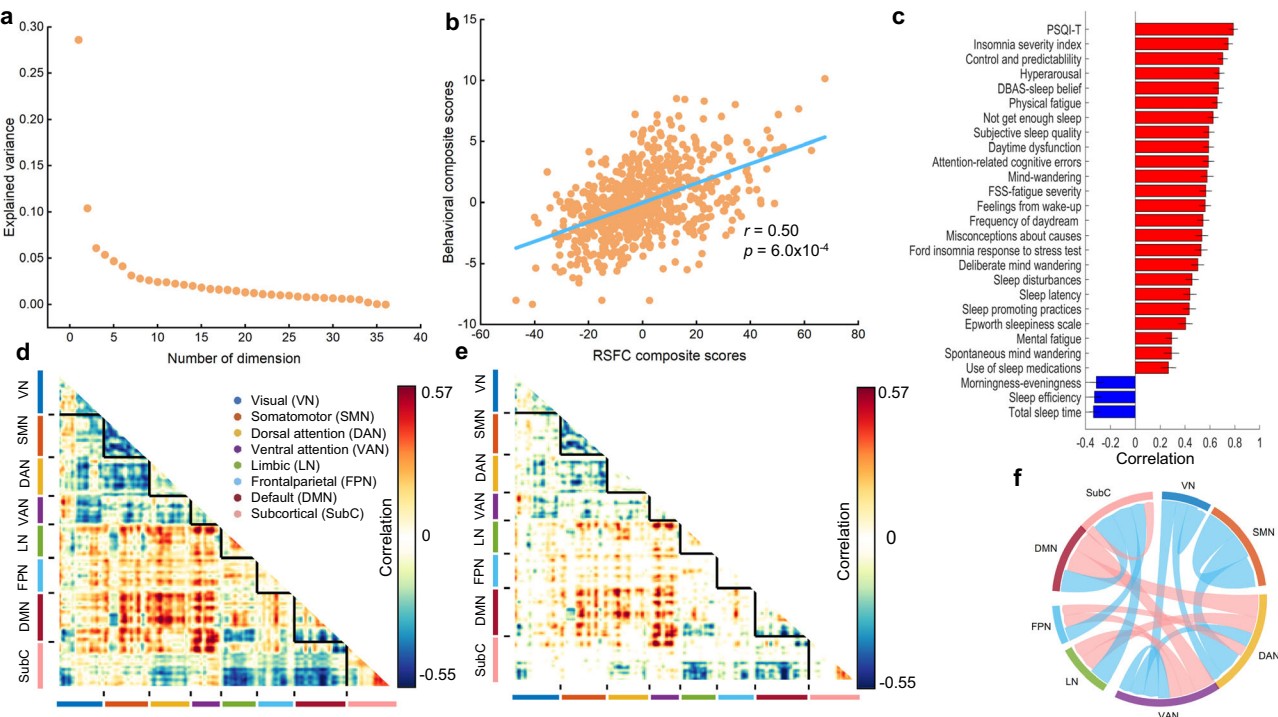

**Fig. 2 | PLS reveals one robust dimension linking sleep health and resting-state function connectivity in the discovery dataset. a** The amount of covariance explained by each latent variable (LV). Each orange dot represents a LV, only the first LV (LV1) survived after permutation testing with FDR correction (*q* < 0.05). This survived dimension (LV1) accounted for 28.6% of resting-state functional connectivity (RSFC)-behavior covariance. **b** scatter plots to illustrate the significant association between individual-specific RSFC and behavioral composite scores of participants in LV1 (*r* = 0.50, two-sided, permuted *p* = 6.0 × 10⁻⁴) using Pearson correlational analysis. **c** significant 27 strongest correlations between participants' behavioral measures and their behavioral composite scores on the group level. Greater loading on LV1 was associated with poorer sleep health. Error bars indicate bootstrapped standard deviation with 1000 bootstrap estimations (*n* = 1000). Behavioral measures for which higher values indicate better sleep health are colored blue. For example, sleep efficiency is colored blue because higher values indicate better sleep health. **d** unthresholded correlations between participants' RSFC data and their RSFC composite scores. Red (or blue) color indicates that greater RSFC is positively (or negatively) associated with LV1. **e** thresholded correlations between participants' RSFC data and their RSFC composite scores (false discovery rate *q* < 0.05). The significant edges were widely distributed throughout the brain, contained a small portion of the total edges in the connectome (5956 edges total out of 30135 or 19.76%). A total of 2666 common edges (8.85% of the 30135 total edges) positively correlated with RSFC composite score and a total of 3290 common edges (10.92% of the 30315 total edges) negatively correlated with RSFC composite score. **f** correlations between participants' RSFC data and their RSFC composite scores, averaged within and between networks defined by Yeo et al's seven networks with significant bootstrapped *Z*-scores. The pink line represents a positive correlation while the blue line represents a negative correlation. DMN, Default mode network; PSQI-T, total score of Pittsburgh Sleep Quality Index. DBAS, Dysfunctional beliefs, and attitudes about sleep scale. FPN Fronto-parietal network, VN Visual network, SMN Somatomotor network, DAN dorsal attention network, VAN ventral attention network, LN limbic network. Source data are provided as a Source Data file.

This survived dimension (LV1) accounted for 28.6% of RSFC-behavior covariance (Fig. 2a), with significant association (*r* = 0.50, permuted *p* = 6.0 × 10⁻⁴) between RSFC and behavioral composite scores (Fig. 2b). Figure 2c shows the top correlations between LV1's behavioral composite/loading score and the 27 significant sleep health measures on the group level. A greater behavioral composite score was associated with poorer sleep health (e.g., poor sleep quality, insomnia, hyperarousal, dysfunctional beliefs and attitudes about sleep, physical fatigue, not getting enough sleep). Correlations between LV1's RSFC composite/loading scores and the RSFC data are shown in Fig. 2d (unthresholded correlations) and Fig. 2e (significant correlations). The significant edges were widely distributed throughout the brain, contained a small portion of the total edges in the connectome (5956 edges total out of 30135 or 19.76%). A total of 2666 common edges (8.85% of the 30135 total edges) positively correlated with RSFC composite score and a total of 3290 common edges (10.92% of the 30315 total edges) negatively correlated with RSFC composite score.

Figure 2f shows the significant RSFC correlations averaged within and between networks defined by Yeo et al's seven network[19]. Greater RSFC composite score was associated with increased RSFC within the subcortical network (SubC) mainly the thalamus (Fig. 3a and Fig. S3h), and increased RSFC between DMN and the dorsal attention network

(DAN), between DMN and the ventral attention network (VAN), between the fronto-parietal network (FPN) and DAN, between FPN and VAN (Fig. 2d, e, f and Fig. S3c, S3d, S3f, S3g). Greater RSFC composite score was associated with decreased RSFC within the Somatomotor network (SMN) and the VAN (Figs. 2d, e, f and 3a; Fig. S3b, S3d), and between the DAN and VAN, between the DAN and SMN, between the SubC and DMN, between the SubC and FPN (Figs. 2d, 2e, 2f, and 3a; Fig. S3c, S3c, S3b, S3h).

Figure 3b further demonstrates the relative importance of the regions in the obtained significant RSFC pattern by showing the top five nodes with highest weighted degree in both the positive and negative networks (Table S3). Specifically, for the positive network, the top five nodes with the greatest number of edges were primarily located in the VAN, i.e., the dorsal insular cortex and precentral gyrus (Fig. 3b). For the negative network, the top five nodes with the greatest number of edges were primarily located in the Subcortical regions, i.e., the thalamus, and somatomotor cortex (e.g., postcentral gyrus and paracentral lobule) (Fig. 3b).

## Cross-dataset replicability and generalizability

The robustness of the obtained LV1 was further ensured with a large replication sample (N = 628) from the BBP (see "Methods" for the

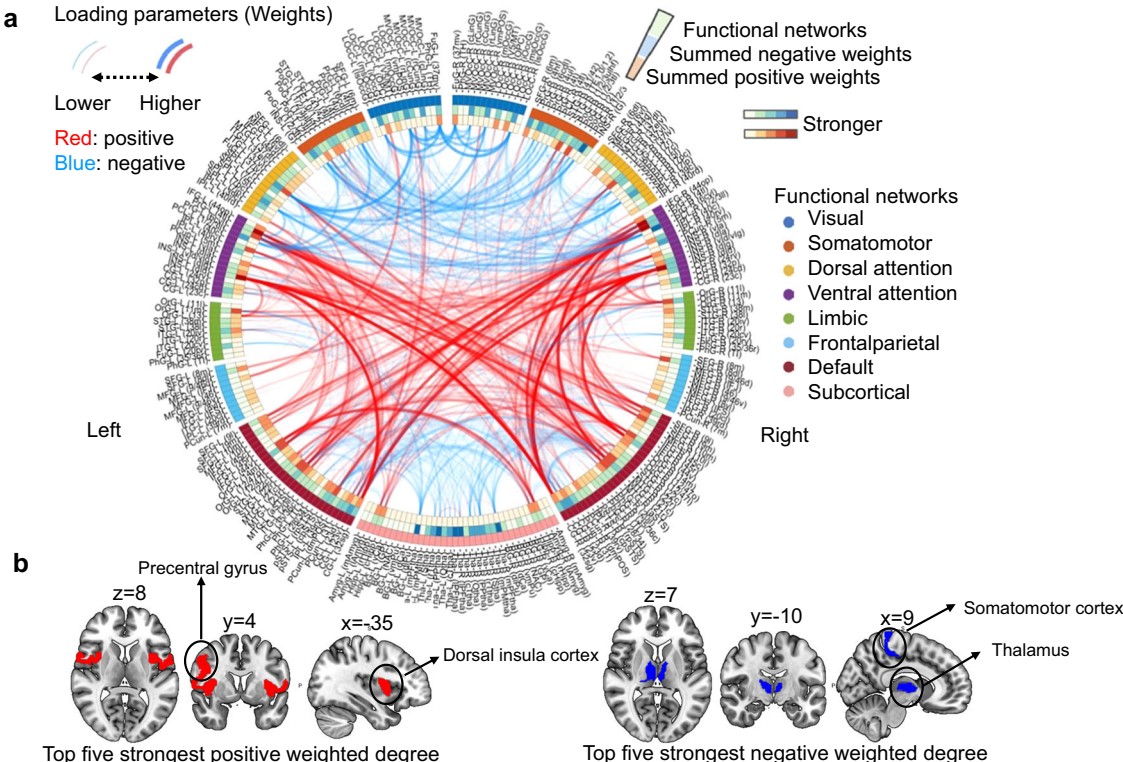

**Fig. 3 | Correlations between LV1's RSFC composite/loading scores and the RSFC data. a** circular plot representation of the correlations between LV1's RSFC composite/loading scores and the RSFC data. From outermost to innermost, the first layer of the circle represents different resting-state networks, and the second and third layers each represent the sum of positive and negative predictive weights coming from each brain region. **b** The top five nodes with the highest degree in both the positive (left side) and negative networks (right side). The abbreviations of the brain regions within each resting-state network can be found on the website: https://atlas.brainnetome.org/download.html. LV, latent variable; RSFC, resting-state functional connectivity. Source data are provided as a Source Data file.

details). To do this, we replicated the PLS procedure conducted in the discovery dataset with the replication dataset (see "Methods" for the details). The LV1 accounted for 26.3% variance and still survived after permutation testing with FDR correction ($q < 0.05$) in the replication dataset with significant association ($r = 0.44$, permuted $p = 0.006$) between RSFC and behavioral composite scores (Fig. S4). Importantly, the obtained LV1 was largely replicated, evidenced by the high correlation between the behavioral salience scores in the discovery and replication dataset ($r = 0.95$, $p ~ = 0$, Fig. 4a), between the behavioral loading scores in the discovery and replication dataset ($r = 0.99$, $p ~ = 0$, Fig. 4c), between the RSFC salience scores in the discovery and replication dataset ($r = 0.71$, $p ~ = . 0$, Fig. 4b), between the RSFC loading scores in the discovery and replication dataset ($r ~ = 0.95$, $p ~ = 0$, Fig. 4d).

We further tested the cross-dataset generalizability by projecting dataset 2 onto the salience parameters learned by PLS in dataset 1. Then, we examined the correlation between the behavioral and RSFC composite scores in dataset 2. We found the obtained LV1 from dataset 1 successfully generalized to dataset 2, as evidenced by a significant correlation between the obtained behavioral and RSFC composite score ($r = 0.25$, permuted $p = 2.66 \times 10^{-6}$, Fig. 4e).

**Spatial correlation with neurotransmitter densities**
Furthermore, we answered question3 (Fig. 1) by testing whether the sleep-health-related connectome (Fig. 2e) was spatially correlated with the distribution of several neurotransmitter systems involved in the domains of sleep health, including serotonin receptors (5-HT1a, 5-HT1b, and 5-HT2a) and transporters (5-HTT), together with metabotropic glutamate receptor 5 (mGluR5) and the γ-Aminobutyric acid type A (GABAA) receptor. First, density values were derived from average group maps of healthy volunteers obtained in prior

multitracer molecular imaging studies. These maps were resampled to an isotropic 2-mm spatial resolution as we did in the fMRI data. Then, we obtained the average value for each region of the Brainconnectome atlas for each Neurotransmitter density map. Next, we summed the positive and negative FC loadings (weighted) separately for each region of the Brainconnectome atlas to represent the region importance scores in positive (Fig. 5a) and negative network (Fig. 5b). Finally, spearman correlation analysis between the region importance score and receptor/transporter densities were conducted[45]. Please refer to the "Methods" and Supplementary Materials for the details. After permutation testing with FDR correction ($q < 0.05$), significant associations were found between on the one hand, the summed positive network and the 5HT1a map (Fig. 6a), the 5HT2a map (Fig. 6b), the GABAA map (Fig. 6c) and the mGluR5 (Fig. 6d) and on the other hand, the summed negative network and the 5HT1a (Fig. 6e) map, and the mGluR5 (Fig. 6f) map. These results suggested the PLS pattern estimated from RSFC may be relevant at the neuromolecular level in line with sleep health.

**Predictive utility of the sleep-health-related connectome: for sleep quality in a healthy population**
Next, considering that the most contributing measure in the LV/dimension found in the discovery ($r = 0.79$, Fig. 2c) and replication ($r = 0.78$, Fig. S4c) BBP samples were the PSQI total score, we further examine the predictive utility of the sleep-health-related connectome features for individual PSQI scores in a completely independent dataset. We here used support vector regression (SVR) to predict the sleep quality measured by the PSQI total score (see Fig. 7a) in the unrelated individuals of the HCP dataset ($n = 435$), see "Methods" for the details. It turned out that the significant edges of the sleep-health-related connectome found in the discovery sample can predict the

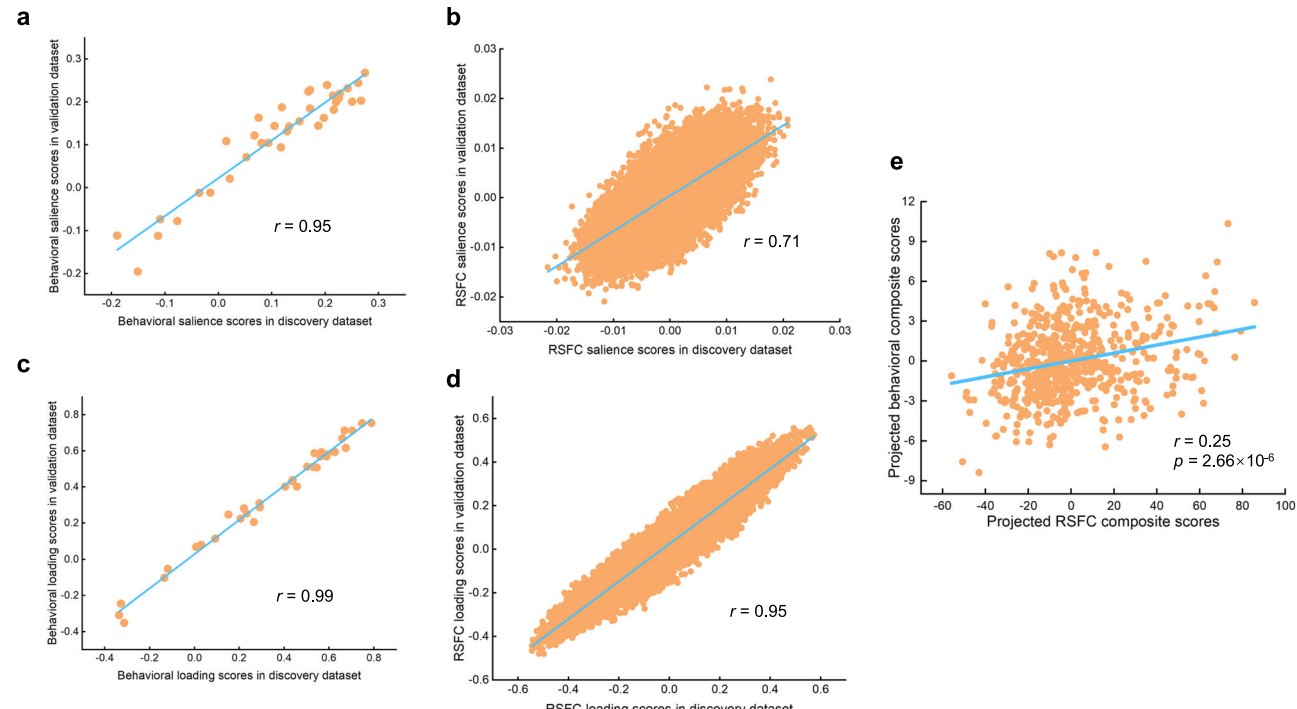

**Fig. 4 | The robustness of the obtained LV1 was further ensured with a large replication dataset. a** High correlation between the behavioral salience scores in the discovery and replication dataset ($r = 0.95$, $p$ - = 0). **b** High correlation between the RSFC salience scores in the discovery and replication dataset ($r = 0.71$, $p$ - = 0). **c** High correlation between the behavioral loading scores in the discovery and replication dataset ($r = 0.99$, $p$ - =0). **d** High correlation between the RSFC loading scores in the discovery and replication dataset ($r$ - = 0.95, $p$ - = 0). **e** The Pearson correlation between the obtained behavioral and RSFC composite score ($r = 0.25$, two-sided, permuted $p = 2.66 \times 10^{-6}$) by projecting dataset 2 onto the salience parameters learned by PLS in dataset 1. LV latent variable, RSFC resting-state functional connectivity, PLS partial least squares. Source data are provided as a Source Data file.

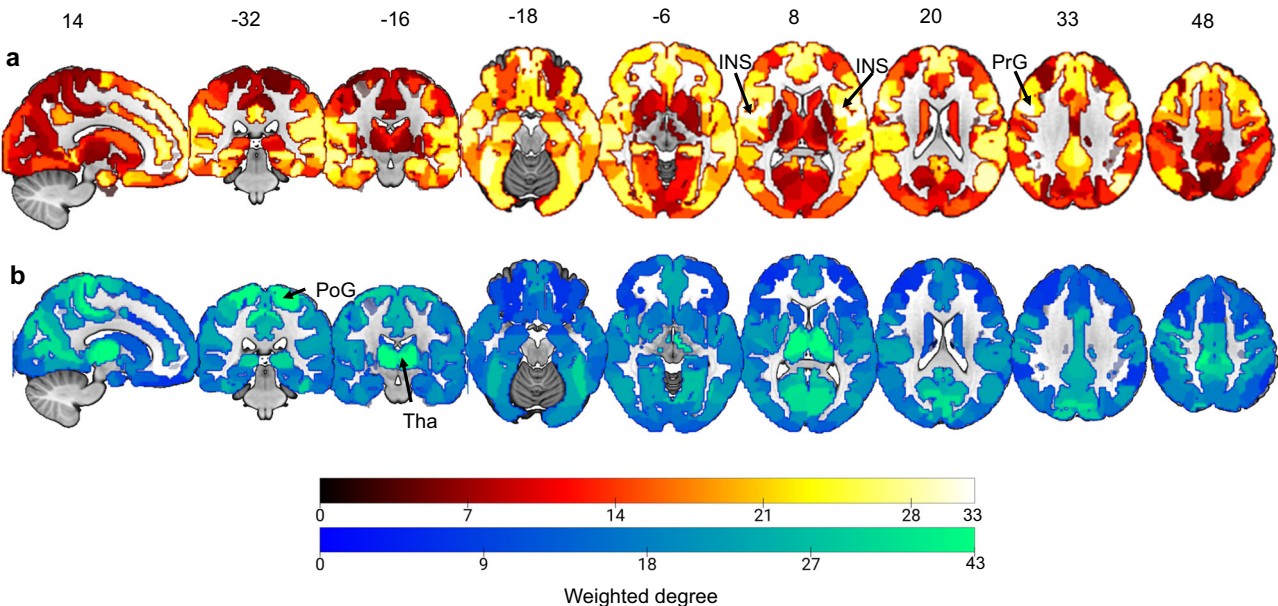

**Fig. 5 | Maps of region importance scores that are covarianced with sleep heath dimension.** The positive and negative RSFC loadings were summed separately for each region of the Brainconnectome atlas to represent the region importance scores in positive (**a**) and negative networks (**b**). RSFC, resting-state functional connectivity. INS, Insular Gyrus; PrG, Precentral Gyrus; PoG, Postcentral Gyrus; Tha, Thalamus. Source data are provided as a Source Data file.

total score of PSQI in the HCP dataset as evidenced by a significant Pearson correlation between the predicted and original PSQI ($r$ average = 0.18, $r$ range = 0.13−0.23 permuted $p < 0.0006$) after a 100 repeated 10-fold cross-validation (Fig. 7b, c). The Mean Absolute Error

(MAE) of the PSQI prediction using the connections identified in the PLS is 2.39. Notably, the predicted PSQI score did not show significant associations with demographic variables and head motion (Table S4), suggesting that the demographic variables and head motion did not

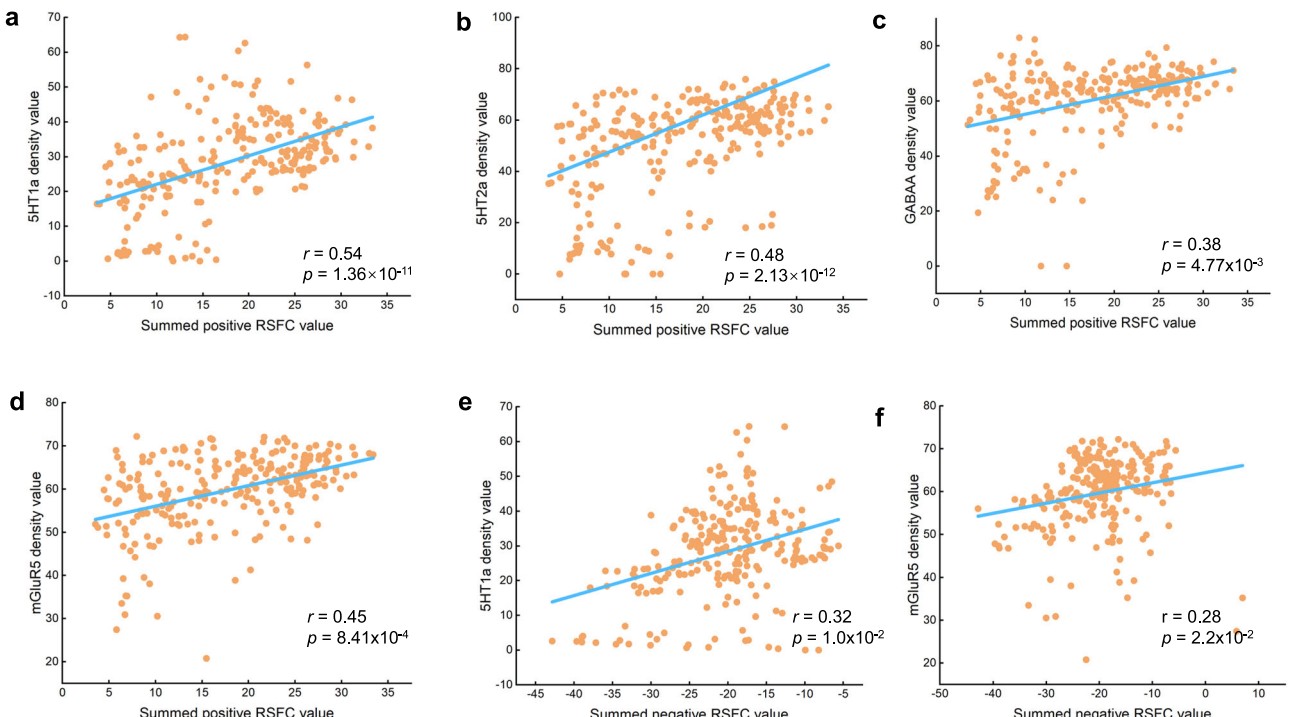

**Fig. 6 | Spatial correlation with neurotransmitter densities.** The RSFC loadings (weighted) were spatially correlated with the distribution of several neurotransmitter systems potentially involved in domains of sleep health with Pearson correlational analysis, specifically, the density value of 5HT1a (**a**, $r = 0.54$, two-sided, $p = 1.36 \times 10^{-11}$), 5HT2a (**b**, $r = 0.48$, two-sided, $p = 2.13 \times 10^{-12}$), GABAA(**c**, $r = 0.38$, two-sided, $p = 4.77 \times 10^{-3}$), mGluR5(**d**, $r = 0.45$, two-sided, $p = 8.41 \times 10^{-4}$) were found to be significantly correlated with the summed positive network; the density value of 5HT1a (**e**, $r = 0.32$, two-sided, $p = 1.0 \times 10^{-2}$), mGluR5 (**f**, $r = 0.28$, two-sided, $p = 2.2 \times 10^{-2}$) were found to be also significantly correlated with the summed negative network. RSFC, resting-state functional connectivity. Source data are provided as a Source Data file.

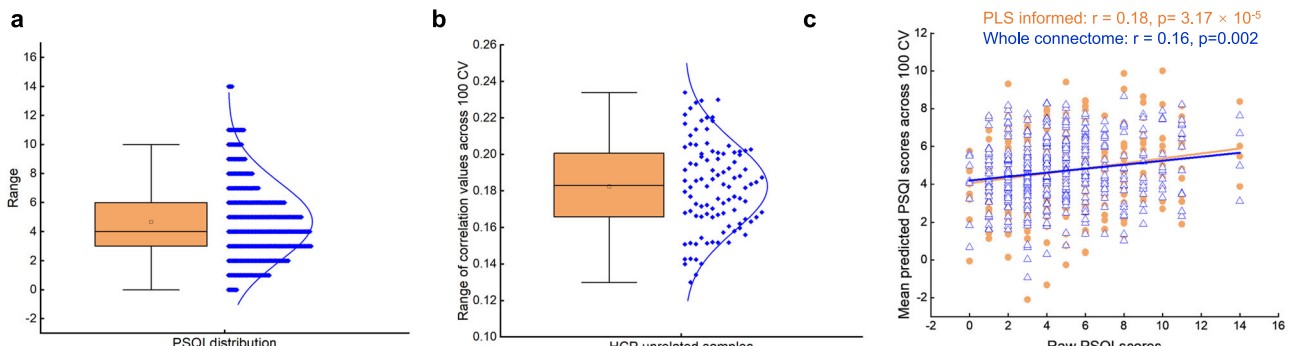

**Fig. 7 | Predictive utility of the sleep-health connectome. a** The distribution of the total score of the Pittsburgh Sleep Quality Index (PSQI) in the Human Connectome Project (HCP) dataset ($n = 435$). For the box plot, minima = 0, maxima = 14, center = 4.68, median = 4, 25th percentile = 3, 75th percentile = 6. **b** The distribution of the correlation value between the predicted (by the RSFC spatial pattern of LV1) and original PSQI after a 100 repeated 10-fold cross-validation ($n = 100$). For the box plot, minima = 0.13, maxima = 0.23, center = 0.18, median = 0.18, 25th percentile = 0.17, 75th percentile = 0.20. **c** The prediction of PSQI total score in the HCP dataset using the significant edges of the RSFC spatial pattern obtained by PLS analysis in dataset 1 as shown in the yellow correlational graph with a Pearson correlation between the original PSQI and the averaged predicted PSQI after a 100 repeated 10-fold cross-validation ($r = 0.18$, two-sided, permuted $p = 3.17 \times 10^{-5}$); the prediction of PSQI total score in the HCP dataset using the whole-brain connectome as depicted in the blue correlational graph with a Pearson correlation between the original PSQI and the averaged predicted PSQI after a 100 repeated 10-fold cross-validation ($r = 0.16$, two-sided, permuted $p = 0.002$). PLS partial least squares, RSFC resting-state functional connectivity, LV latent variable. Source data are provided as a Source Data file.

significantly affect the prediction. Importantly, we also examined the prediction performance of models using the whole-brain connectome. Our results here showed that models using the specific sleep-health-related connectome yield similar prediction performance with whole-brain connectome models (after a 100 repeated 10-fold cross-validation, $r$ average = 0.16, $r$ range = 0.10–0.22, permuted $p = 0.002$, MAE = 2.33 (the blue correlational graph in Fig. 7c)) hence demonstrating that the identified sleep-health-related connectome captures sufficient and relevant information for individual prediction of sleep phenotype in the adult population.

## Predictive utility of the sleep-health-related connectome: for sleep disorders

Furthermore, we investigated the possibility of distinguishing insomnia patients from sleep-healthy controls based on the sleep-health-related connectome. To accomplish this, we utilized a Gaussian radial

**a**

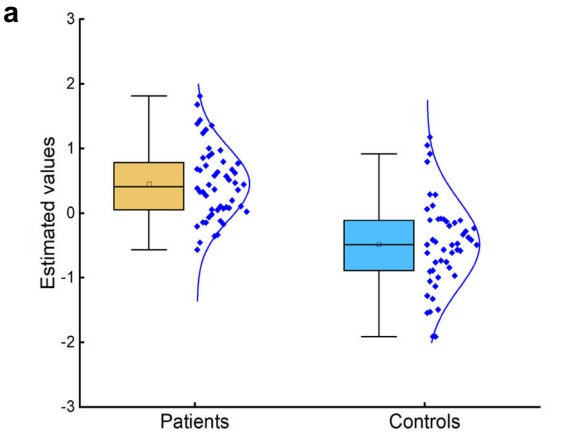

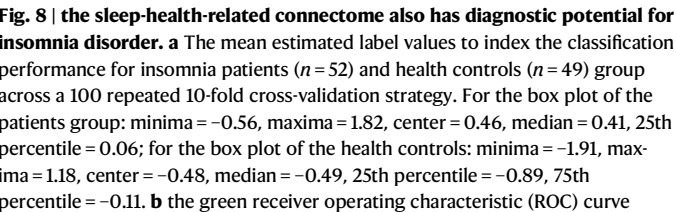

**b**

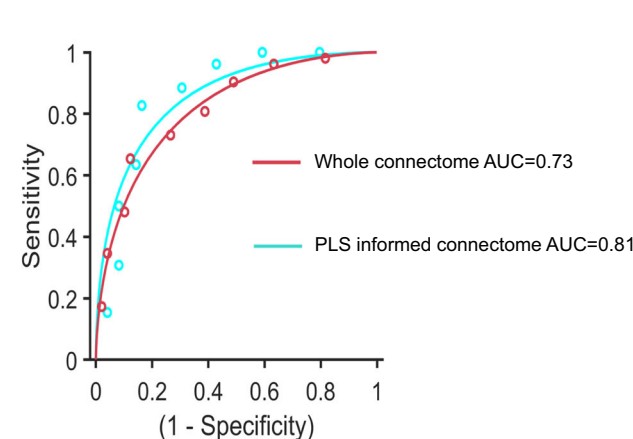

**Fig. 8 | the sleep-health-related connectome also has diagnostic potential for insomnia disorder. a** The mean estimated label values to index the classification performance for insomnia patients ($n = 52$) and health controls ($n = 49$) group across a 100 repeated 10-fold cross-validation strategy. For the box plot of the patients group: minima = −0.56, maxima = 1.82, center = 0.46, median = 0.41, 25th percentile = 0.06; for the box plot of the health controls: minima = −1.91, maxima = 1.18, center = −0.48, median = −0.49, 25th percentile = −0.89, 75th percentile = −0.11. **b** the green receiver operating characteristic (ROC) curve depicts the classification performance using the significant edges of the RSFC spatial pattern obtained by PLS analysis in dataset4, suggesting the sleep-health-related connectome also has diagnostic potential to distinguish insomnia patients from sleep-healthy subjects with an average accuracy across 100 CV of 79.73%, permuted $p = 2.39 \times 10^{-3}$; the red ROC curve depicts the classification performance using whole-brain connectome with an average accuracy of 72.62%, permuted $p = 7.53 \times 10^{-3}$. LV latent variable, RSFC resting-state functional connectivity. Source data are provided as a Source Data file.

basis function (RBF) kernel support vector machine (SVM) classifier which was implemented using the LIBSVM toolbox[46] on our classification dataset (dataset4) containing 52 insomnia patients and 49 sleep-healthy subjects (see "Methods" and Table S6 for the sample details). The classifier performance was validated using a 100-repeated 10-fold cross-validation strategy (see "Methods" for details). Classification performance that had an accuracy higher than 70% was considered to be meaningful. It turned out that the sleep-health-related connectome also has diagnostic potential to distinguish insomnia patients from sleep-healthy subjects in dataset4 with an average accuracy across 100 CV of 79.73% (permuted $p = 2.39 \times 10^{-3}$) with 78.85% sensitivity, 83.67% specificity and area under the curve (AUC) = 0.81 (Fig. 8a, green receiver operating characteristic (ROC) curve in Fig. 8b). Importantly, we tested to which extend the sleep-health-related connectome does as well or better than using the whole-brain connectome for the classification performance. It turned out the whole-brain connectome-based classification performance was lower as indicated by an average accuracy across 100 CV of 72.62% (permuted $p = 7.53 \times 10^{-3}$) with 78.85% sensitivity, 67.35% specificity and AUC = 0.73 (red ROC curve in Fig. 8b).

Moreover, the classification performance based on the sleep-health-related connectome can also be generalized to an external classification dataset containing 35 insomnia patients and sleep-healthy subjects with an average accuracy across 100 CV of 78.29% (permuted $p = 3.89 \times 10^{-3}$) with 80.00% sensitivity, 77.14% specificity and area under the curve (AUC) = 0.79 (Supplementary Fig. S5a, green receiver operating characteristic (ROC) curve in Fig. S5b). Sample details about this external classification dataset can be found in Supplementary Methods. We also tested to which extent the sleep-health-related connectome does as well or better than using the whole-brain connectome for the classification performance in the external classification dataset. It turned out the whole-brain connectome-based classification performance was lower as indicated by an average accuracy across 100 CV of 74.27% (permuted $p = 6.27 \times 10^{-3}$) with 74.29% sensitivity, 77.14% specificity and AUC = 0.75 (red ROC curve in Fig. S5b). These results together suggest that the sleep-health-related connectome feature sets have specific classification power for sleep

health phenotype at the individual level and could therefore be seen as useful biomarkers.

## Discussion

There is growing interest toward using a conceptual framework that articulates sleep health as multi-facet[21,47]. Although the importance of RSFC in multi-facets of sleep health is increasingly recognized, the degree to which RSFC is associated with sleep health as a multivariate dimension remains largely unknown. Leveraging a unique dataset including resting-state fMRI and behavioral assessments spanning multiple domains of sleep health, we found robust correlated patterns of RSFC and sleep health measures that could be represented in one dimension. The derived pattern estimated from RSFC shows neurobiological relevance and predictive utility for sleep phenotype in healthy populations. Critically, the RSFC spatial pattern of this dimension additionally shows diagnostic potential to distinguish insomnia patients from sleep-healthy subjects. Together these results delineate the RSFC-guided dimension of sleep health, which could serve as a foundation to develop a reliable brain connectome biomarker for sleep health with high potential for clinical translation-and ultimately for the diagnosis, prognosis, treatment, and prevention of sleep health-related problems.

In the present study, the measurement of sleep health composite was guided by both behavior and brain functional connectivity. The obtained dimension of sleep health was associated with a unique cluster of 27 out of the 36 selected SH measures mainly including sleep quality, the severity of insomnia, degree of hyperarousal, degree of dysfunctional beliefs and attitudes about sleep, degree of physical fatigue, and degree of subjective sleep lack/deprivation, etc. (Fig. 2c). This significant cluster of SH variables were still fallen within the proposed seven domains and largely overlapped with the domains of sleep health examined in previous studies[17,36] but draw a multifaceted picture of SH to a bigger extend based on its broader definition[2]. The present study thus can be regarded as a step forward in measuring and quantifying sleep health composite by incorporating whole-brain resting-state functional connectivity[3]. To this end, the present study validated the theory that sleep health can be operated and constructed as a

composite dimension and that sleep health domains do not exist in isolation[9,12].

Notably, four out of six measures in the domain of beliefs, attitudes, and habits about sleep[48] were found to be important measurements in the significant latent dimension, with the measure of diminished control and predictability contributing the most. Existing evidence suggests that individuals with more positive beliefs and attitudes about sleep are more likely to experience better sleep health[37,49]. On the other hand, previous studies have proposed that the dysfunctional beliefs about sleep (e.g., diminished control and predictability of sleep) can result in the misperception of sleep duration (e.g., underestimate of sleep duration)[50], especially in people who are suffering from insomnia[23,51]. One possible explanation is that dysfunctional beliefs may alter the constructive process involved in remembering the amount of sleep obtained, thereby contributing to misperception[51]. Also recent theories claimed that people tend to gather evidence that confirms our beliefs rather than evidence that challenges them[52], so it is crucial to disprove the dysfunctional beliefs to improve sleep health. In correspondence, Cognitive Behavioral Therapy for Insomnia (CBT-I) emphasized the importance of cognitive restructuring and sleep hygiene to alter the dysfunctional beliefs, attitudes, and habits about sleep as its two main components[53]. Taken together, future studies investigating sleep health should pay more attention to the domain of beliefs, attitudes, and habits about sleep.

The latent dimension of sleep health was associated with a unique spatial pattern of resting-state functional connectivity, namely a positive and negative network pattern. For the positive network pattern, poorer sleep health was associated with hyper-connectivity between the network involved in internally oriented attention (DMN) and the network responsible for modulation of the goal-directed attention (DAN), between DMN and the network involved in processing of salience (VAN), between the network involved in external goal-directed regulation (FPN) and DAN, between FPN and VAN as well as hyper-connectivity within the subcortical network (SubC) mainly the thalamus. The heightened coupling between the higher-order and attentional networks may suggest a decreasing dedifferentiation of the high-order system to the middle attentional system[19,54] while the increased FC mainly within the thalamus (a brain system responsible for gating information) further indicates sustained ascending arousal input from the thalamus[16] in poor sleep health individual phenotype. Notably, the positive network consisted of the highest-degree nodes in the left dorsal insular cortex and precentral gyrus. As key hubs of the salience network, these brain regions play a crucial role in cognitive and emotional processes[23,55]. Impaired connectivity patterns between these areas and other regions could underlie cognitive, vigilance, and perception dysfunctions, as well as subjective distress and sleep complaints[56].

For the negative network pattern, poorer sleep health was also associated with hypo-connectivity between the two attentional networks, between DMN and SubC mainly the thalamus, between FPN and SubC mainly the thalamus, between DAN and the network involved in sensory and motor perception (SMN), as well as hypo-connectivity within the VAN and SMN. The reduced coupling between and within the attentional networks and the SMN may mainly suggest the collapse of top-down orienting of attention, bottom-up salience processing[57,58] and sensory and motor perception[22] while the reduced connectivity between the high-order networks and thalamus further reflects the reduction in the upward transmission of information to the higher-order networks by the thalamus as a relay station[59] in the condition of poor sleep health. Moreover, the negative network consisted of the highest-degree nodes in the thalamus and somatomotor cortex. Both the thalamus[59] and the somatomotor cortex[22] play a crucial role in sleep physiology. Impaired connectivity patterns between these areas and other regions could suggest the dysfunction of relaying sensory and motor signals to the cerebral cortex and regulating sleep, alertness, and consciousness.

Both the positive and negative network patterns were spatially correlated with the distribution of several neurotransmitter systems involved in sleep health, including the serotonin receptors, the glutamate receptor (mGluR5), and the GABAA receptor. Previous studies have proven that: (1) serotonin functions predominantly to promote wakefulness and to inhibit REM (rapid eye movement) sleep (REMS)[31]; (2) γ-aminobutyric acid (GABA) is increasingly recognized as an important inhibitory neurotransmitter for the initiation and maintenance of sleep[30,60]; (3) glutamate is the primary excitatory neurotransmitter in the central nervous system, its altered levels were found in people who suffered from insomnia disorders[23,32,61]. Taken together, the linked distribution of these neurotransmitter systems and the RSFC spatial pattern found in the present study further suggest that the derived pattern estimated from RSFC is neurobiologically relevant with regard to neuromolecular systems engaged in sleep health.

Critically, the RSFC spatial pattern of this dimension also has diagnostic potential to distinguish insomnia patients from sleep-healthy subjects with an accuracy of 79.73% and 78.29% in dataset4 and the external classification dataset respectively. While a previous study utilized a "resting-state" fMRI support vector machine (SVM) classifier that achieved higher classification accuracy between IDs and HCs, it included 5 min of wakefulness, as well as data from sleep stages 1 (S1), 2 (S2), and 3 (S3) for each subject[62]. However, this approach could not exclude the possibility of shared individualized characteristics across multiple sessions from the same subject, which could inflate the classification performance. Importantly, our classification results demonstrated that the obtained brain connectome pattern also contained pathophysiological relevance for insomnia disorder. Insomnia is a common, distressing, and clinically complex symptom, that causes difficulty initiating sleep; frequent awakening; or early-morning awakening with daytime dysfunction[23,32]. Despite its high prevalence, insomnia often goes under-diagnosed and untreated, resulting in general fatigue and decreased productivity[63]. Based on the RSFC spatial pattern-based SVM classifier derived from the obtained sleep health dimension, we will be able to aid in the diagnosis of insomnia disorder. To this end, these findings may suggest that insomnia is a condition with a multifaceted pathophysiology spanning over deficits in multiple domains of sleep health. Ultimately, the present study can serve as a foundation to identify targets that may enable sleep-based interventions for improving both brain health and clinical outcomes.

The strength of the present study includes the utilization of a large sample, advanced multivariate methods, and replication of results in an independent sample. The component we identified with RSFC not only showed macromolecular relevance but also had diagnostic potential, generalized well to unrelated individuals in the HCP dataset, and was robust across alternative methodological strategies. Nonetheless, our work has several limitations. First, here we sought to delineate linked dimensions of sleep health and intrinsic brain functional connectivity, uncovering dimensions of sleep health that are guided by and linked to underlying brain connectome properties. Although we included the largest sleep health domains in the neuroimaging studies of sleep health, to the best of our knowledge, this approach necessarily is limited by the available variables measuring sleep health in the BBP, which did not represent the full domains of sleep health. Future studies can incorporate more sleep-related variables (i.e., sleep regularity) to depict the sleep health dimension. Second, our current analysis only considered functional connectivity and behavioral measures of sleep health. Future research could incorporate rich multi-modal imaging data, objective sleep measurements with actigraphy and polysomnography, biochemics, and genomics. Third, the present study only used subjective measures to control for the state of alertness of individuals during the resting-state fMRI acquisition. Future research could use more objective measurements such as EEG, camera, or eye-tracking to monitor alertness/drowsiness states during resting-state fMRI scanning. Fourth, the sample mainly

consisted of young adults in pattern recognition and middle-aged adults in the prediction and classification test, leaving the RSFC-guided sleep health dimension in other age groups such as children, adolescents, and the elderly unknown. Recently, the definition of what constitutes good sleep health was adapted to pediatrics[64] as well as the elderly[65]. Hence, it is of great importance for future studies to also examine the RSFC-informed sleep health dimension in both younger and older populations. Fifth, it should be noted that causality between sleep health and its associated RSFC component cannot be established in this cross-sectional study. Future studies can use a longitudinal design to test whether worse sleep health leads to alterations of this sleep-health-related connectome.

In summary, in this study, we discovered and replicated multivariate patterns of intrinsic functional connectivity that are highly correlated with a dimension of sleep health in a large sample of young adults. This dimension was composed of a unique cluster of sleep health and unique features of resting-state functional connectivity, namely a positive and negative network pattern. Both the positive and negative network patterns were spatially correlated with the distribution of several neurotransmitter systems involved in sleep health. Moreover, the obtained positive and negative network patterns also contained pathophysiological relevance for insomnia disorder. These results thus delineate the connectivity-guided dimension of sleep health, which could serve as a foundation for developing brain connectome-based biomarkers in sleep symptomatology.

## Methods

### Participants

**Discovery and replication datasets (dataset 1 and 2).** The main dataset comprises individuals from the Behavioral Brain Research Project of Chinese Personality (BBP)[66], which was launched in September 2019 (still in progress) to recruit participants from that year's freshmen at Southwest University, Chongqing, China. In total, 1369 participants completed the cross-sectional neuroimaging protocol as well as the behavioral measures of sleep health. To create two independent samples for discovery and replication analyses, we split the participants according to their data collection periods (September–December 2019 [discovery dataset], namely freshman enrolled in the year 2019; September–December 2020 [replication dataset] and freshman enrolled in the year 2020). Specifically, a discovery dataset ($n = 712$) and a replication dataset ($n = 657$) were created. Of the discovery dataset, 25 participants were excluded due to excessive head motion during scanning (e.g., with a mean framewise displacement [FD] larger than 0.3 mm), resulting in a final sample of 687 participants (mean age 18.96, SD = 0.95; 233 males and 454 females). Applying the same exclusion criteria to the replication dataset produced 628 participants (mean age 19.16, SD = 1.03; 208 males and 420 females). See Table 1 for detailed demographics of each dataset. The two datasets were confirmed to also have similar basic demographic variables, i.e., age, sex, and race (Table 1), as well as head motion (Table 1). Participants were compensated for 50 Chinese Yuan for their participation. Full informed consent from each participant was obtained by BBP Consortium, and research procedures and ethical guidelines were followed in compliance with Southwest University (SWU) institutional review board approval.

**HCP external validation dataset (dataset 3).** The dataset 3 was selected from the 1200 Subjects Release of the Human Connectome Project (HCP). This dataset provides high-quality behavioral/demographic and imaging data from healthy young adults (https://www.humanconnectome.org/)[67]. From the original HCP dataset, we first included a group of participants ($n = 1084$) for which L-R resting-state fMRI data as well as PSQI scores were available. From this group, we further excluded data from 46 participants based on the exclusion criteria indicated as follows: (1) participants with missing values on

## Table 1 | Demographic characteristics of participants in the discovery and replication dataset

|  | Discovery dataset (N = 687) | Replication dataset (N = 628) |
|---|---|---|
| Age, mean (SD), years | 18.96(0.95) | 19.16(1.03) |
| Female | 454(66.08) | 420(66.88) |
| Handedness-Right | 606(88.21) | 575(91.56) |
| BMI, mean (SD) | 21.23(2.87) | 21.85(3.19) |
| **Race** | | |
| Han | 569(82.82) | 527(83.92) |
| Other | 118(17.18) | 101(16.08) |
| Framewise displacement (FD), mean (SD) | 0.10(0.05) | 0.10(0.05) |
| Total brain volume, mean (SD), mm³ | 1541331.47(138185.45) | 1539323.87(158514.00) |
| **Family income per year (yuan, RMB)** | | |
| <5000 | 125(18.20%) | 93(14.81%) |
| 5001 – 25000 | 277(40.32%) | 262(41.72%) |
| 25001 – 45000 | 108(15.72%) | 96(15.29%) |
| 45001 – 65000 | 55(8.01%) | 66(10.51%) |
| 65001 – 85000 | 39(5.68%) | 36(5.73%) |
| 85001 – 105000 | 45(6.55%) | 31(4.94%) |
| >105,000 | 38(5.53%) | 44(7.01%) |
| PANAS-P, mean (SD) | 28.96(6.12) | 29.45(6.20) |
| PANAS-N, mean (SD) | 17.66(5.97) | 17.39(6.00) |

*PANAS* Positive and negative affect schedule, *SD* standard deviation.

demographic variables such as age, sex, education, BMI, and race or family information; (2) participants with a history of hyper/hypothyroidism or history of other endocrine problems; (3) women who had recently given birth; and (4) participants having a mean FD more than 0.3 mm. Importantly, to exclude the influence of shared genetic and environmental factors, we randomly kept one subject from each family[68], resulting in 435 final unrelated subjects. See Table S5 for detailed demographics of each sample. Full informed consent from each participant was obtained by the Washington University–University of Minnesota (WU–Minn) HCP Consortium, and research procedures and ethical guidelines were followed in compliance with WU institutional review board approval.

**Classification dataset (dataset4).** The classification dataset involved 109 participants, including 56 patients with insomnia disorder (ID) and 53 healthy controls (HC). Part of the dataset was collected from the Sleep Center, Department of Brain Disease of Chongqing Traditional Chinese Medicine Hospital (CTCMH), which included 25 ID and 36 HC participants (referred as the CTCMH dataset), and the remaining part of the dataset is from the Sleep and Neuroimaging Center, Southwest University (referred as SNIC dataset). All the participants received the MRI scanning at the brain imaging center of Southwest University. Patients with insomnia disorder were diagnosed by experienced hospital psychiatrists (D.G. and CY.L.) according to the International Classification of Sleep Disorders: Diagnostic and Coding Manual, 3rd ed., and insomnia symptoms have lasted at least three nights a week for more than 3 months. All the health controls met the following criteria: a good sleep habit and a good sleep onset and/or maintenance; a regular dietary habit; no consumption of any stimulants, medications, alcohol, and cocaine for at least 3 months before the study; no history of neurological or psychological disorders; lower scores of Pittsburg Sleep Quality Index (PSQI) than 7[69]. Participants with any findings of pathological brain MRI as well as ineligibility for MRI scanning (any type of metal implant) were excluded from the

study. Eight participants were excluded due to excessive head motion during the scanning (e.g., with a mean FD larger than 0.3 mm), resulting in a final sample of 52 ID patients (mean age = 44.00 years old, SD = 12.39) and 49 HC (mean age = 42.10 years old, SD = 15.93). Participants were compensated for 80 Chinese Yuan for their participation. The research projects were approved by the SWU and CTCMH institutional review boards, and written informed consent was obtained from each participant in accordance with the Declaration of Helsinki. The two groups were matched for age, sex, and head motion (Table S6). The participants' demographic characteristics are summarized in Table S6.

### Sleep health-related assessments in the discovery and replication dataset

A large set of sleep health measures (Table 2 and Table S1) were included in the current study based on the consideration to select

available variables in the Behavioral Brain Research Project of Chinese Personality (BBP) that 1) represent central domains of SH described in the Ru SATED[3] as well as in the National Sleep Foundation (NSF)'s Sleep Health Index[6]; 2) provide high consistency with previous SH studies[15,70]; 3) draw a multifaceted picture of SH to a bigger extend based on its broader definition by including variables that are sleep deficiency related, nighttime- and daytime-related, quantitative and qualitative informed. On that basis, the chosen 36 variables were grouped into seven domains: (1) Satisfaction with sleep/sleep quality (e.g., PSQI total score, Not get enough sleep, Feelings from wake-up, Necessity of nap, Needed nap time, Subjective sleep quality); (2) Alertness during waking hours (e.g., Epworth sleepiness scale, Mind-wandering, Spontaneous mind wandering, Deliberate mind wandering, Frequency of daydream, Fatigue severity, Physical fatigue, Mental fatigue, Attention-related cognitive errors, Valid sleep cue reaction time(RT), Invalid sleep cue RT, Valid sleep cue accuracy (ACC), Invalid

**Table 2 | Mean and standard deviation (SD) of the sleep health measures in the discovery and replication dataset**

| Domains of sleep health | Measures | Discovery dataset (N = 687) Mean (SD) | Replication dataset (N = 628) Mean (SD) |
|---|---|---|---|
| Satisfaction with sleep/ Sleep quality | Not get enough sleep | 2.11 (0.74) | 2.07 (0.75) |
| | Feelings from wake-up | 2.35 (0.82) | 2.32 (0.76) |
| | Necessity of nap | 3.09 (0.89) | 2.98 (0.91) |
| | Needed nap time | 4.22 (1.14) | 4.19 (1.15) |
| | PSQI-T (otal score) | 4.97 (2.44) | 5.33 (2.58) |
| | Subjective sleep quality | 1.16 (0.83) | 1.10 (0.84) |
| Alertness during waking hours | Epworth sleepiness scale | 8.91 (3.38) | 8.92 (3.47) |
| | Mind-wandering | 13.06 (3.67) | 12.84 (3.77) |
| | Spontaneous mind wandering | 19.11 (4.88) | 18.36 (4.80) |
| | Deliberate mind wandering | 18.11 (4.23) | 17.85 (4.49) |
| | Attention-related cognitive errors | 30.32 (6.80) | 30.60 (7.87) |
| | Frequency of daydream | 33.19 (9.07) | 31.28 (8.78) |
| | Fatigue severity | 38.96 (8.53) | 38.89 (8.88) |
| | Physical fatigue | 4.21 (2.41) | 3.95 (2.53) |
| | Mental fatigue | 2.63 (1.17) | 2.66 (1.22) |
| | Valid sleep cue RT | 515.85 (71.61) | 521.43 (83.61) |
| | Invalid sleep cue RT | 538.14 (76.47) | 553.40 (90.26) |
| | Valid sleep cue ACC | 0.95 (0.06) | 0.95 (0.06) |
| | Invalid sleep cue ACC | 0.94 (0.07) | 0.94 (0.07) |
| Timing of sleep | Morningness-eveningness questionnaire | 13.60 (2.84) | 13.22 (2.84) |
| Sleep efficiency/ continuity | Sleep efficiency | 0.94 (0.06) | 0.93 (0.07) |
| | Sleep latency | 0.68 (0.79) | 0.73 (0.80) |
| | Wake-up times | 0.30 (0.69) | 0.36 (0.80) |
| Sleep duration | Total sleep time | 6.74 (0.87) | 6.86 (0.94) |
| Sleep deficiency | Insomnia Severity Index | 7.60 (4.25) | 7.47 (4.31) |
| | Hyperarousal | 31.90 (9.50) | 32.12 (9.72) |
| | Ford insomnia response to stress test | 20.42 (5.30) | 20.78 (5.26) |
| | Sleep disturbances | 0.86 (0.50) | 0.89 (0.49) |
| | Use of sleep medications | 0.03 (0.243) | 0.06 (0.35) |
| | Daytime dysfunction | 1.38 (0.82) | 1.75 (0.83) |
| Sleep beliefs, attitudes, and habits | DBAS-total score | 128.74 (33.44) | 129.75 (32.84) |
| | Misconceptions about causes | 5.73 (1.72) | 5.79 (1.66) |
| | Diminished control and predictability | 3.42 (1.43) | 3.43 (1.47) |
| | Unrealistic sleep expectations | 4.67 (1.63) | 4.68 (1.72) |
| | Misattribution of the consequences | 3.02 (1.59) | 3.13 (1.68) |
| | Faulty beliefs about sleep-promoting practices | 4.00 (1.26) | 4.03 (1.28) |

*RT* reaction time, *ACC*, accuracy, *DBAS* Dysfunctional Beliefs and Attitudes about Sleep Scale.

sleep cue ACC); (3) Timing of sleep (e.g., Morningness-eveningness questionnaire); (4) Sleep efficiency/ continuity (e.g., Sleep efficiency, Sleep latency, Wake-up times); (5) Sleep duration (e.g., Total sleep time); (6) Sleep deficiency (e.g., Insomnia severity index, Hyperarousal, Ford insomnia response to stress test, Sleep disturbances, Use of sleep medications, Daytime dysfunction); (7) Sleep beliefs, attitudes, and habits (e.g., Dysfunctional beliefs and attitudes about sleep scale (DBAS)-total score, Misconceptions about causes, Diminished control and predictability, Unrealistic sleep expectations, Misattribution of the consequences, Faulty beliefs about sleep promoting practices). For the specific descriptions about the chosen variables, see Table 2 and Supplementary Table S1.

### Data acquisition and image preprocessing
#### The discovery and replication sample
**Data acquisition.** All MR Images were acquired using a 3 T Siemens Primsa-fit scanner with a standard 32-channel head coil located at Southwest University. A high-resolution T1-weighted structural image was obtained using a three-dimensional gradient sequence in order to facilitate alignment of individual subject images into a common space (repetition time (TR) = 2530 ms, time of echo (TE) = 2.98 ms, field of view (FOV) = 256 × 256 mm$^2$, thickness = 1 mm, voxel size = 0.5 × 0.5 × 1 mm$^3$, flip angle = 7°, resolution matrix = 256 × 256, slices = 192, slice oversampling = 33.3%, phase-encoding direction = AC » PC). Approximately 8 min of rs-fMRI data containing 240 volumes were acquired for each subject using a blood oxygen level-dependent (BOLD-weighted) sequence (TR = 2000 ms; TE = 30 ms; slices = 62; slice thickness = 2 mm; FOV = 224 × 224 mm$^2$; flip angle = 90°; resolution matrix = 112 × 112; voxel size = 2 × 2 × 2 mm$^3$; phase-encoding direction = PC » AC). In order to minimize motion, prior to data acquisition participants' heads were stabilized in the head coil using one foam pad over each ear and a third over the top of the head. During the resting-state scanning, a fixation cross was displayed as images were acquired. Participants were instructed to stay awake, keep their eyes open, fixate on the displayed crosshair, and remain still.

**Image preprocessing.** The preprocessing steps on all collected neuroimaging data were performed using the publicly available CONN functional connectivity toolbox (version 20.b; https://www.nitrc.org/projects/conn), together with SPM12 (Wellcome Department of Cognitive Neurology, London, UK; http://www.fil.ion.ucl.ac.uk/spm). Functional images were (1) slice time corrected, (2) underwent motion correction and susceptibility artifact correction based on field map, (3) warped into Montreal Neurological Institute(MNI) standard space using the diffeomorphic Anatomical Registration Through exponentiated Lie Algebra (Dartel) approach to realign the 3D anatomical data into Montreal Neurological Institute space[71], (4) smoothed spatially with a Gaussian kernel of 6 mm full width at half maximum (FWHM). Next, functional images further underwent denoising steps using the anatomical component-based correction (aCompCor) method[70]. Specifically, noise signals, including signals from cerebrospinal fluid and white matter (WM) (five principal components), and movement parameters (six motion parameters, six temporal derivatives, and their squares), and linear trend were removed from the images as confounds[72]. Subsequently, data scrubbing was implemented to address head motion concerns. The bad time points were regarded as regressors defined as volumes with FD power > 0.5 mm as well as the two succeeding volumes and one preceding volume to reduce the spillover effect of head motion[72]. Finally, functional images were filtered using a bandpass filter (0.008–0.09 Hz) to reduce the effects of very low-frequency drifts and high-frequency noises.

#### HCP sample
**Data acquisition.** Structural (T1 and T2 images, required for preprocessing functional neuroimaging data) and functional MRI data were collected at Washington University on the Siemens 3 T Connectome Skyra scanner using a multi-band sequence. The structural images were 0.7 mm isotropic. The rs-fMRI data were 2 mm isotropic with TR = 0.72 s. Full details of the acquisition parameters for the HCP data can be found elsewhere[73]. Two sessions of rs-fMRI data were collected on consecutive days for each subject, and each session consisted of one or two runs. The length of each rs-fMRI scan was 14.4 min (1200 frames). Here, the analyses were restricted to individuals for whom the left–right phase-encoding scans for the rs-fMRI session were completed and available. For rs-fMRI data acquisition, participants were asked to lie with eyes open, with a "relaxed" fixation on a white cross (on a dark background), think of nothing in particular, and not fall asleep. Details of the data collection can be found elsewhere[74]. Details about behavioral measures including PSQI can be found in HCP S1200 Data Dictionary(https://db.humanconnectome.org/data/projects/HCP_1200) and[75].

#### Image preprocessing
We adopted the preprocessed data provided by the Human Connectome Project (HCP S1200 release), for which also the spatial normalization to the MNI152 template had already been performed before download. Details of the structural and functional data preprocessing can be found in the HCP S1200 manual[76], and we used version 3.21 of the HCP preprocessing pipeline. Consistent with the BBP dataset, the HCP downloaded functional images were further smoothed, and underwent regression of motion and non-relevant signals, including linear trend, Friston 24 head motion parameters, white matter (CompCor, 5 principal components), and CSF signal (CompCor, 5 principal components), scrubbed and filtered.

#### Classification sample
**Data acquisition.** All MR Images in the SNIC dataset were acquired using a 3 T scanner (Magnetom TIM- Trio, Siemens, Erlangen, Germany) with a standard head coil located at Southwest University. In order to minimize motion, prior to data acquisition participants' heads were stabilized in the head coil using one foam pad over each ear and a third over the top of the head. High- resolution T1- weighted anatomical images were collected using 3D spoiled gradient recalled (3DSPGR) sequence (TR/TE = 8.5/3.4 ms, flip angle = 12°, resolution matrix = 512 × 512, FOV = 240 × 240 mm$^2$, with a voxel size of 1 × 1 × 1 mm$^3$, 176 slices, 1 mm thickness). Approximately 5 min of rs-fMRI data containing 204 volumes were acquired for each subject using an echo-planar imaging (EPI) sequence (TR/TE = 1500/29 ms, flip angle = 90°, resolution matrix = 64 × 64, voxel size = 3 × 3 × 3 mm$^3$, FOV = 192 × 192 mm$^2$, axial slices = 25, thickness/ gap = 5/0.5 mm). The first four volumes were discarded to ensure steady-state longitudinal magnetization. During the resting-state scanning, participants were instructed to fix on a crosshair in the center of the black background screen without thinking intentionally in the mind and keep as motionless as possible[77].

All MR Images in the CTCMH dataset were acquired by using the same scanner as the discovery and replication dataset. The scanning parameters are also the same. During the resting-state scanning, all participants were instructed to fix on a crosshair in the center of black background screen without thinking intentionally in the mind and keep as motionless as possible.

#### Image preprocessing
The preprocessing steps on all collected neuroimaging data in the classification sample were performed using fMRIPrep 21.0.1[78,79]

(RRID:SCR_016216), which is based on Nipype 1.6.1[80](RRID:SCR_002502). Details of the structural and functional data preprocessing can be found in the supplementary materials.

## Resting-state functional connectivity construction

To construct RSFC, the preprocessed data in the discovery and replication dataset were first parcellated using the Brainnetome Atlas[42], which includes 210 cortical regions and 36 subcortical regions. For each participant, the BOLD time course of each node was extracted by taking the mean across voxels. Pearson correlation coefficient ($r$) between the time courses of each pair of nodes was calculated. A Fisher's r-to-z transformation was performed to improve the normality of the correlation coefficients, which resulted in a 246 × 246 symmetric functional connectivity matrix with 30,135 [(245 × 246)/2] edges for each participant.

## Partial least squares analysis

We used PLS analysis to examine the relationship between resting-state functional connectivity (RSFC) and sleep health (SH) measures (Fig. 1) in the discovery dataset. PLS analysis is a multivariate statistical technique that derives latent variables (LVs), by finding weighted patterns of variables from two given datasets that maximally covary with each other[36,81]. In the present analysis, 1 variable set corresponded to RSFC and the other to behavioral measures spanning multiple domains of sleep health. The two variable sets were correlated with each other across participants, and the resulting covariance matrix was subjected to singular value decomposition to identify the latent brain-behavior dimensions. Specifically, each LV is comprised of an RSFC pattern at the node level ("RSFC saliences") and an SH profile ("SH saliences"). Individual-specific RSFC and SH composite scores for each LV were obtained by linearly projecting the RSFC and SH measures of each participant onto their respective saliences. See the Supplementary Methods section "Partial least squares analysis" for mathematical details. Before the PLS analysis, we regressed out the confounding effects from both RSFC and behavior data including mean FD, age, handedness, and sex.

Inference and validation of the statistical model were performed using nonparametric methods including: (1) statistical significance of overall patterns was assessed by permutation tests using 1000 permutations for behavioral data; (2) the importance (measured as loading scores) of feature (RSFC, sleep health measures) was evaluated by bootstrap resampling; (3) the generalizability of each LV was assessed by 10-fold cross-validation with 200 repetitions. Mathematical details of the analysis and inferential methods are described in the Supplementary Methods section "Partial least squares analysis" and results. False discovery rate (FDR) correction ($q < 0.05$) was applied to all analyses.

## Control and reliability analyses

We further tested whether LVs obtained in the discovery dataset were robust to (1) global signal regression, (2) total intracranial volume (including gray matter, white matter, and cerebrospinal fluid) regression, (3) time (hour) of acquisition regression, (4) the pre-scanning positive and negative affect regression, (5) BMI regression, (6) family income regression, (7) adding confounding variables (age, sex, handedness and head motion) into the phenotypic data for the PLS analysis, (8) non-Gaussian distributions of the behavioral data with quantile normalization, as well as (9) using a different brain (Seitzman et al.'s) Atlas[43] containing 300 regions for the RSFC construction. To assess the robustness of each LV, we computed Pearson's correlations between RSFC (or SH) saliences obtained in each of the eight-control analysis and RSFC (or SH) saliences from the original PLS analysis. Please refer to (able S2 for the results.

## Internal validation

As an internal validation of the obtained LVs or dimensions in the discovery dataset, we first replicated the PLS analysis described in the section "Partial least squares analysis", and "Control and Reliability Analyses" with the replication dataset. Then, Pearson's correlation coefficient between the behavioral salience scores in the discovery and replication dataset, between the behavioral loading scores in the discovery and replication dataset, between the RSFC salience scores in the discovery and replication dataset, between the RSFC loading scores in the discovery and replication dataset were computed respectively.

We further tested the cross-dataset generalizability by projecting the dataset 2 onto the salience parameters learned by the PLS analysis in dataset 1. Then, we examined the correlation between the behavioral and RSFC composite scores in dataset 2. The significance of the correlation value was further assessed by a permutation test (behavioral data of dataset 2 shuffled 1000 times).

## External prediction validation

Considering that the PSQI total score contributes the most in the LV/ dimension found in the discovery BBP sample ($r = 0.79$), we further tested whether the RSFC spatial pattern of this dimension observed in the discovery dataset can predict the sleep quality of unrelated individuals in the HCP dataset as an external prediction validation (Fig. 1). In particular, we adopted the support vector regression (SVR), the most widely used algorithm in multivariate neuroimaging research, to examine the predictive performance of the significant RSFC pattern in the LV found in the discovery sample on sleep quality in the HCP dataset. Specifically, we used all the edges of the significant RSFC pattern as the features. The 10-fold cross-validation was used to determine the optimal parameter C for SVR. Specifically, the optimal parameter was finally selected according to the prediction performance after the inner 10-fold cross-validation was performed for each parameter ([0.1,1] with a step of 0.1) in turn and was further utilized in the outer loop to accomplish the final prediction. All subjects were divided into 10 subsets by 10-fold cross-validation (outer loop). Nine of them were used as training sets and model fitting was performed. The fitted model parameters and the remaining one subset were used as the test set to generate predicted values. This process was repeated 10 times to generate predicted values for all subjects. In order to obtain stable predictive performance, we repeated the above prediction pipeline 100 times to generate 100 predicted scores for each participant and further averaged these predicted scores to obtain robust estimates. Pearson's correlation coefficient between the predicted average and actual values was computed to provide final estimates of predictive performance. The significance was evaluated by randomly shuffling the PSQI values 1000 times and running the above prediction pipeline for each time to obtain a null distribution of correlation coefficients between the predicted and actual values.

## Spatial correlation with neurotransmitter densities

We further test whether the FC loadings from the PLS analysis were spatially correlated with the distribution of several neurotransmitter systems potentially involved in domains of sleep health. Based on the literature[31,32,60,61,82], serotonin receptors (5-HT1A, 5-HT1B, and 5-HT2A) and transporters (serotonin reuptake transporter [5-HTT]), together with metabotropic glutamate receptor 5 (mGluR5) and the γ-Aminobutyric acid type A (GABAA) receptor were investigated. Density values were derived from average group maps of healthy volunteers (5-HT1a: $n = 35$; 5-HT1b: $n = 36$; 5-HT2a: $n = 29$; 5-HTT: $n = 100$; mGluR5: $n = 22$; GABAA: $n = 11$) obtained in prior multitracer molecular imaging studies[33] (details were provided in the Table S7). These maps were resampled to an isotropic 2-mm spatial resolution as we did in the fMRI data. Then, we obtained the average value of each region of the Brainconnectome atlas for each Neurotransmitter density map, i.e., a 246x1 matrix. Further, we summed the positive and negative FC loadings separately for each region of the Brainconnectome atlas to represent the region importance scores in positive (Fig. 6a) and

negative network (Fig. 6b). Finally, we conducted a Spearman correlation analysis between the region importance score and receptor/transporter densities calculated for these regions[45]. To establish the statistical significance of a spatial correlation against chance, we conducted spatial permutation tests to obtain a null distribution of correlation coefficients for region importance scores and neurotransmitter densities extracted from 5000 permutations while accounting for the spatial autocorrelation of brain regions[83,84]. The p-value was determined by empirically observed spatial similarity values compared to the null distribution. The significance level was set at FDR corrected $p < 0.05$.

### Clinical diagnosis validation

We further evaluated the possibility of distinguishing the insomnia patients ($n = 52$) from sleep-healthy controls ($n = 49$) in the classification dataset based on the RSFC spatial pattern of the found LV/dimension (details about the classification sample were given in the section of participants). Specifically, we used all the edges of the significant RSFC pattern as the features. To accomplish this, we utilized a Gaussian radial basis function (RBF) kernel SVM classifier which was implemented using the LIBSVM toolbox[46]. We used the recommended exponentially growing sequences of parameters of C and γ by LIBSVM group, i.e., $C$ ($2^{-5}$, $2^{-3}$,...,$2^{15}$) and $\gamma$ ($2^{-15}$,$2^{-13}$,...,$2^{3}$) based on a practical guide to SVM[85]. The classifier performance was validated using a 10-fold cross-validation strategy. Specifically, all subjects were divided into 10 subsets by 10-fold cross-validation. Nine of them were used as training sets and model fitting was performed. The fitted model parameters and the remaining one subset were used as the test set to generate classified labels. This process was repeated 10 times in sequence to generate classified labels for all subjects, and then calculate the classification accuracy. Regarding the optimal values for the RBF kernel parameters $C$ and $\gamma$, they were tuned based on the training set using a grid search strategy based on a 10-fold cross-validation (inner loop). Then the SVM classifier was trained by applying the optimized parameters to the training set, based on which the testing set was finally classified. In order to obtain stable classification performance, we repeated the above pipeline 100 times to generate 100 classification performance values and further averaged these values to obtain robust estimates.

Statistical significance was evaluated by using 1000 permutation tests with a threshold of $p < 0.05$. In Brief, labels of participants were randomly shuffled 1000 times and split into the training and testing sets. Classification performance that had an accuracy of higher than 70% was considered to be meaningful. Moreover, we replicated the classification procedure conducted in the classification dataset with an external classification dataset to distinguish insomnia patients ($n = 35$) from sleep-healthy controls ($n = 35$), please refer to Supplementary Methods for more details.

### Reporting summary

Further information on research design is available in the Nature Portfolio Reporting Summary linked to this article.

## Data availability

The HCP consortium database used in this study was freely available at the following accessible link: https://db.humanconnectome.org/. The BBP sample data are available under restricted access for it is still an ongoing project, access can be obtained by contacting a joint team (PIs: Q.H.H., J.Q., T.Y.F., H.C., and X.L.). The significant FC weights from the BBP discovery cohort are available at GitHub: https://github.com/wangyulinatUGent/Sleep_Health_Dimension. The classification sample used in this study is available upon request from the corresponding author. The PET/SPECT data from the prior vivo molecular imaging studies[33] used in this study are available at the following accessible link:

https://github.com/netneurolab/hansen_receptors/tree/main. Source data are provided in this paper.

## Code availability

We used the Matlab code from https://github.com/danizoeller/myPLS[86], based on Krishnan and associates' work[35] to implement the PLS calculation. The code for spatial permutation testing can be found at https://github.com/frantisekvasa/rotate_parcellation. The codes for SVR and SVM analysis are openly available at https://www.csie.ntu.edu.tw/~cjlin/libsvm/[46]. The brain maps were presented using the MRIcroGL toolbox (https://www.nitrc.org/projects/mricrogl). The connectome maps were presented using code from (https://github.com/cocoanlab/cocoanCORE/tree/master/Visualization)[87].

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

## Acknowledgements

This work has been funded by the National Key Research and Development Program of China (2021YFC2501500, X.L.), the National Natural Science Foundation of China (31971028, X.L.; 82202247, D.B.D.; 32300861, Y.L.W.), and 2021 International Exchange Program and Introduction Project Funding awarded to Y.L.W.

## Author contributions

Y.L.W. conceived and implemented the study, carried out the data analysis, and wrote the manuscript. X.L. supervised the work. S.G. supported the data analysis, especially the multivariate methods, and reviewed and edited the manuscript. D.B.D. supported the calculation of the spatial correlation between the sleep-health-related connectome and the distribution of neurotransmitter systems. Both D.B.D. and F.Z. helped with the machine-learning methods and commented on the manuscript. C.Y.L. acquired and shared dataset4. K.Y. and D.H.Y acquired and shared the external classification dataset. Q.H.H., J.Q., T.Y.F., H.C., and X.L. funded, designed, and implemented the Behavioral Brain Research Project to Chinese Personality (BBP) project.

## Competing interests

The authors declare no competing interests.
