## [Peer Review File · Nature Communications]

Covariance patterns between sleep health domains and distributed intrinsic functional connectivityReviewer #1 (Remarks to the Author):

This study investigates relationships between latent components of resting-state functional connectivity (RSFC) and sleep health related variables using partial least squares methods in large independent samples. RSFC components related to sleep health were then shown related to the presence of insomnia in another, clinical, sample. The paper is well written and the findings are interesting. However a major weakness found in many RSFC papers is the lack of control for circadian timing of acquisitions and most importantly the state of alertness of individuals during the resting state fMRI acquisition. This aspect is highly critical as it has been shown (and also reviewed by the authors) that sleepiness in the scanner, which often happens with resting-state acquisitions where participants are simply instructed to look at a cross or even close their eyes, has profound influence on RSFC patterns. This is very important for studies attempting to link sleep disorders or variables with RSFC as the absence of control for alertness level prevents meaningful conclusions to be reached (are these effects related to the sleep variable in question or the reflection of drowsiness during the resting state?).

At the very least fMRI bouts with shifts in alertness (as evidenced by EEG or camera or eye-tracking) should be removed from the analyses, and time (hour) of acquisition (if not consistent across participants) should be included as covariate. Unfortunately this information is often not available in large cohorts not designed for sleep-related analyses.

Reviewer #2 (Remarks to the Author):

Reviewer #2 Attachment on the following page

Comments to authors:

In this paper, Wang et al. used the partial least squares, a multivariate data-driven approach to identify the covariance patterns between multiple-sleep health domains and distributed intrinsic functional connectivity. With the increasing recognition of resting state functional connectivity in sleep health, an important question arises as can we identify a composite sleep health dimension that relates to RSFC patterns? This study serves as one of the first attempts to address this question, and therefore the results are novel. Due to our modern around- the-clock lifestyle, high work pressure, and psychosocial stress, many people experience poor sleep health on a regular basis, to this end, this paper is highly likely to have broad readership. Major strengths of this work include the utilization of a large unique sample containing the largest sleep health domains in the neuroimaging studies of sleep health, advanced multivariate methods, and the exhaustive investigation of result robustness across different datasets, data processing pipelines and potential confounds. The component the authors identified with RSFC not only show macromolecular relevance, but also had diagnostic potential, generalized well to unrelated individuals in the HCP dataset.

Overall, I feel the manuscript is well-motivated, clearly written, the analyses to be exquisitely conducted. I only have a few concerns that would like the authors to address prior to publication.

1. In the introduction(page4), the authors stated that “Also in line with these considerations, the high dependence and interaction across different domains of sleep health encourages a transition in sleep health research¹⁴ from categorical to dimensional approaches that integrate intrinsic brain functional connectivity and sleep health.”, can the authors elaborate more on “high dependence and interaction across different domains of sleep health”? Otherwise, as a reader who is not specialized in sleep research, he/she can be confused.
2. I am very fond of the idea to seek the neuromolecular relevance of the brain connectome underlying interindividual differences in sleep phenotype. However, the motivation to look into the sleep-health related neurotransmitters is not so straightforward to me, though the authors stated that they aimed to find the neuromolecular relevance of the sleep health related brain connectome. I highly recommend the authors to add more information regarding why they choose to calculate the spatial correlation between the sleep-health related connectome and the distribution of several neurotransmitter systems based on the neurotransmitter receptor maps.
3. While the authors regressed out the head motion parameters, I would recommend the authors to check whether the yielded sleep health behavior composite is correlated with head motion parameters, which would help to exclude the possible influence of head motion.

4. Although combining RSFC and behavior has advantages and yielded meaningful patterns, I would recommend the authors to use other approaches, such as principal component analysis (PCA) to sleep health behavioral measures to test whether the first latent component was robust across analytical approaches.
5. In addition, there are two popular approaches, i.e., CCA and PLS, to implement multivariate covariance. Although there was no standard to choose which one is better for a given situation, the authors may want to simply state their considerations.
6. Furthermore, while many previous studies used the traditional PLS, some studies advocate the application of dimensionality reduction and/or regularization constraints when performing PLS. The authors opted not to perform any dimensionality reduction or to add any regularization constraint. The authors may need to make their consideration clear.
7. The strength of the present study includes the utilization of a large sample, advanced multivariate methods, and replication of results in an independent sample. A relative weaker point of the present study, in my opinion, is the relatively small sample size of the classification dataset with 52 insomnia patient and 49 healthy controls. Although the component identified with RSFC not only show macromolecular relevance, but also generalized well to unrelated individuals in the HCP dataset and were robust across alternative methodological strategies, a burning question is whether the component identified with RSFC can differentiate insomnia patients from healthy controls in another independent dataset, in other words, can the classification performance of the sleep-health related connectome generalized to another independent dataset? If the classification performance based on the sleep-health related connectome can also be generalized to another classification dataset, then we can say the results are robust and the RSFC pattern derived from the PLS analysis can be regarded as biomarker for sleep health.
8. I am also recommended to discuss the classification performance of the sleep-health related connectome findings (page15) in more depth by diving deeply into the research literature about insomnia classification based on a machine learning approach to separate subjects with and without insomnia symptoms using the RSFC features.

Minor

9. In Fig 5., it can help the reader if the authors can point out the brain regions with high importance scores in both the positive and negative networks.
10. Considering this study is based on the cross-section design, the limitation of casual

interface should be acknowledged.

Reviewer #3 (Remarks to the Author):

This study used a large and repeated sample to identify a sleep health-related brain connectivity pattern using a multivariate approach, PLS. The results showed that a composite sleep health dimension (LV1) was reflected by the variations in connectivity patterns, and this finding was replicated in two other datasets. In addition, the identified sleep-health related brain connectome showed possibility for diagnosis of insomnia disorder.

This is an interesting paper that improves our understanding of the link between sleep-health and brain connectivity. I have the following comments and suggestions for the authors:

1. Although the sleep-health component LV1 can interpret 28.6% of RSFC-sleep covariance, whether this component can be the best to represent for sleep-health. I suggest the authors to provide evidence, such as whether the brain connectome found by LV1 is actually had higher accuracy than by PSQI to identify insomnia disorder?
2. The results of the correlation with neurotransmitter densities might also require validation analysis to show its specificity. For example, the authors could provide evidence that this sleep-related connectome is related to sleep-related neurotransmitter densities, but not to emotion-related or cognitive-related neurotransmitter maps.
3. The accuracy for distinguishing insomnia patients is less than 80%. I suggest the authors test the top ten strongest weighted degree, five positive and five negative, which is also more likely to have clinical translation potential.
4. The discovery dataset and the replication dataset are separated by one year. Please clarify if there is a subject replication or if all participants are new.
5. The number of permutation and bootstrap tests for the PLS analysis was 1000 and 500, respectively. However, considering the large number of participants and connections, the number of tests used here seems relatively small. It is recommended to increase the number of tests to improve the credibility of the results.
6. In the PLS analysis, the z-score was first converted to a p-value to identify significant connections, and then FDR was used to correct for multiple comparisons. However, because PLS analysis is computed in a single analytical step, corrections for multiple comparisons are not required. Instead, the z-score threshold can usually be set to identify significant connections. Therefore, the analysis steps used here do not seem to reflect the advantages of PLS analysis. Please clarify the consideration of FDR correction here.
7. In the external validation of the prediction, the performances of the PSQI prediction using all the connections and the connections identified in PLS were 0.16 and 0.18, respectively. If the authors want to conclude that the sleep-related connections performed BETTER, please add statistical evidence, such as the z-test between these two rs. Also, there do not seem to be any indicators similar to the MAE reported in the manuscript, so it is impossible to know how much the actual value deviates from the predicted value.
8. Please add the range of parameters C and γ used in clinical diagnostic validation.
9. For Table 1, the sleep characteristics should be reported, instead of family income per year.
10. It is not necessary to mention nutrition at the beginning of the introduction.

**Response to reviews:
NCOMMS-23-02175-T**

REVIEWER COMMENTS

Reviewer #1 (Remarks to the Author):

Reviewer comment 1.1

This study investigates relationships between latent components of resting-state functional connectivity (RSFC) and sleep health related variables using partial least squares methods in large independent samples. RSFC components related to sleep health were then shown related to the presence of insomnia in another, clinical, sample. The paper is well written and the findings are interesting.

Author reply 1.1. We are glad that the reviewer found our paper is well-written and the findings are interesting.

Reviewer comment 1.2

However, a major weakness found in many RSFC papers is the lack of control for circadian timing of acquisitions and most importantly the state of alertness of individuals during the resting state fMRI acquisition. This aspect is highly critical as it has been shown (and also reviewed by the authors) that sleepiness in the scanner, which often happens with resting-state acquisitions where participants are simply instructed to look at a cross or even close their eyes, has profound influence on RSFC patterns. This is very important for studies attempting to link sleep disorders or variables with RSFC as the absence of control for alertness level prevents meaningful conclusions to be reached (are these effects related to the sleep variable in question or the reflection of drowsiness during the resting state?). At the very least fMRI bouts with shifts in alertness (as evidenced by EEG or camera or eye-tracking) should be removed from the analyses, and time (hour) of acquisition (if not consistent across participants) should be included as covariate. Unfortunately, this information is often not available in large cohorts not designed for sleep-related analyses.

Author reply 1.2. We express our gratitude to the reviewer for bringing our attention to the importance of controlling for the circadian timing of acquisitions and the state of alertness of individuals during the resting state fMRI acquisition, as these factors can confound RSFC studies. When we designed the Behavioral Brain Research Project to Chinese Personality (BBP), we were already aware of the significance of monitoring the participants' alertness or sleepiness during their resting-state scanning. To address this, we administered the Amsterdam Resting-State questionnaire (ARSQ) 2.0 to the participants after they underwent the resting-state fMRI scanning, as described by Alexander Diaz et al. in 2014¹. The ARSQ 2.0 evaluates participants' mind wandering and includes a dimension called "Sleepiness" which consists of three questions: "1. I felt tired; 2. I felt sleepy; 3. I had difficulty staying awake."

To demonstrate that the sleep-health dimension was independent of drowsiness during the resting state, we extracted the dimension score of "Sleepiness" from the ARSQ 2.0 and correlated it with both the RSFC and behavior composite scores obtained from the discovery dataset. The results revealed no significant correlation between participants' sleepiness and either their RSFC composite score ($r = -0.0349$, $p = 0.3605$) or their behavior composite score ($r = -0.0223$, $p = 0.5603$). Additionally, we examined the acquisition time to confirm that the sleep-health related connectome was independent of the circadian timing of acquisitions. The analysis showed no significant relationship between the acquisition time and either the RSFC composite score ($r = 0.0395$, $p = 0.3013$) or the behavior composite score ($r = -0.0393$, $p = 0.3941$). Furthermore, we followed the reviewer's suggestion to include the time (hour) of acquisition of the resting-state scanning as covariates. The correlation between the saliences of the original PLS and the PLS with corresponding variable regression was high ($r = 0.99$, Table S2), indicating that the PLS components remained robust to the regression of the acquisition time. Collectively, these findings suggest that the sleep-health dimension was related to the sleep variable in question and not merely a reflection of drowsiness during the resting state. Accordingly, we clarified this point in the revised manuscript (page 8) as:

“Sixth, to demonstrate the sleep health component was independent of both the circadian timing of acquisitions and the state of alertness of individuals during the resting state fMRI acquisition, we extracted the dimension score of "Sleepiness" from the Amsterdam Resting-State questionnaire (ARSQ)2.0¹ and time (hour) of acquisition, then correlated them with both the RSFC and behavior composite scores for the discovery dataset. The results revealed no significant correlation between participants' sleepiness and either their RSFC composite score ($r = -0.0349$, $p = 0.3605$) or their behavior composite score ($r = -0.0223$, $p = 0.5603$). Additionally, no significant relationship between the acquisition time and either the RSFC composite score ($r = 0.0395$, $p = 0.3013$) or the behavior composite score ($r = -0.0393$, $p = 0.3941$) were observed.”

However, it is important to acknowledge a limitation of the present study, namely the lack of more objective measurements such as EEG, camera, or eye-tracking during the resting-state fMRI scanning in the BBP project to control for the state of alertness of individuals during the resting-state fMRI acquisition. So, we added this point as a limitation in the revised manuscript (page 18) as:

“Third, the present study only used subjective measures to control for the state of alertness of individuals during the resting-state fMRI acquisition. Future research could use more objective measurements such as EEG, camera, or eye-tracking to monitor alertness/drowsiness states during resting-state fMRI scanning.”

Reviewer #2(Remarks to the Author):

In this paper, Wang et al. used the partial least squares, a multivariate data-driven approach to identify the covariance patterns between multiple-sleep health domains and distributed intrinsic functional connectivity. With the increasing recognition of resting state functional connectivity in sleep health, an important question arises as can we

identify a composite sleep health dimension that relates to RSFC patterns? This study serves as one of the first attempts to address this question, and therefore the results are novel. Due to our modern around- the-clock lifestyle, high work pressure, and psychosocial stress, many people experience poor sleep health on a regular basis, to this end, this paper is highly likely to have broad readership. Major strengths of this work include the utilization of a large unique sample containing the largest sleep health domains in the neuroimaging studies of sleep health, advanced multivariate methods, and the exhaustive investigation of result robustness across different datasets, data processing pipelines and potential confounds. The component the authors identified with RSFC not only show macromolecular relevance, but also had diagnostic potential, generalized well to unrelated individuals in the HCP dataset.

Overall, I feel the manuscript is well-motivated, clearly written, the analyses to be exquisitely conducted. I only have a few concerns that would like the authors to address prior to publication.

Author reply. We are very happy that the reviewer found the results are novel and the manuscript is well-motivated, clearly written, the analyses to be exquisitely conducted. We also appreciate the reviewer's careful reading of the manuscript and suggestions for improving the manuscript.

Reviewer comment 2.1

In the introduction(page4), the authors stated that “Also in line with these considerations, the high dependence and interaction across different domains of sleep health encourages a transition in sleep health research¹⁴ from categorical to dimensional approaches that integrate intrinsic brain functional connectivity and sleep health.”, can the authors elaborate more on “high dependence and interaction across different domains of sleep health”? Otherwise, as a reader who is not specialized in sleep research, he/she can be confused.

Author reply 2.1: We thank the reviewer for prompting us to clarify on “high dependence and interaction across different domains of sleep health”. In the revised manuscript, we clarified this point by adding “(e.g., short sleep duration is usually accompanied with lower efficiency and regularity, lower sleep satisfaction/quality)”.

Reviewer comment 2.2

I am very fond of the idea to seek the neuromolecular relevance of the brain connectome underlying interindividual differences in sleep phenotype. However, the motivation to look into the sleep-health related neurotransmitters is not so straightforward to me, though the authors stated that they aimed to find the neuromolecular relevance of the sleep health related brain connectome. I highly recommend the authors to add more information regarding why they choose to calculate the spatial correlation between the sleep-health related connectome and the distribution of several neurotransmitter systems based on the neurotransmitter receptor maps.

Author reply 2.2: Following the reviewer's recommendation, we have added more information regarding why we choose to calculate the spatial correlation between the sleep-health related connectome and the distribution of several neurotransmitter systems based on the receptor maps in the revised manuscript (page 5), detailed as:

“A growing number of studies highlights the relationships between alterations of some neurotransmitter systems including the serotonin receptors, glutamate and γ -aminobutyric acid (GABA) and sleep disturbances or "unhealthy" sleep²³⁴. Accordingly, we here examined whether the brain connectome underlying interindividual differences in sleep phenotype may be related to specific neurotransmitter systems (based on neurotransmitter receptor maps⁵).”

Reviewer comment 2.3

While the authors regressed out the head motion parameters, I would recommend the authors to check whether the yielded sleep health behavior composite is correlated with head motion parameters, which would help to exclude the possible influence of head motion.

Author reply 2.3: We totally understood the reviewer's concern here and have checked whether the yielded sleep health behavior composite is correlated with head motion parameters. Given the regression of head motion parameters was performed prior to the PLS analysis, there was no correlation between head motion and behavior or RSFC composite scores ($r=0$). To demonstrate the impact of head motion, we conducted PLS again without regressing out the head motion parameters. We found that head motion was not correlated with either the behavior composite scores ($r=0.0320$, $p=0.4031$) or the RSFC composite scores ($r=0.0367$, $p=0.3373$).

Reviewer comment 2.4

Although combining RSFC and behavior has advantages and yielded meaningful patterns, I would recommend the authors to use other approaches, such as principal component analysis (PCA) to sleep health behavioral measures to test whether the first latent component was robust across analytical approaches.

Author reply 2.4: We followed the reviewer's recommendation to perform PCA to sleep health behavioral measures, then computed the Pearson's correlations between the first principal component and the behavioral saliences of LV1 identified with the PLS analysis. It turned out that the first principal component of the sleep health behavioral measures was highly correlated with the behavioral saliences of the LV1, $r=0.91$, $p=1.22 \times 10^{-14}$. This high correlation thus indicated that the first latent component was robust across analytical approaches. We added this control analysis in the revised manuscript (page8) as:

“Seventh, the first latent component was robust across analytical approaches, which was evidenced by the high correlation between the first principal component of the sleep health behavioral measures

(obtained by principal component analysis) and the behavioral saliences of LV1($r=0.91$, $p=1.22 \times 10^{-14}$).”

Reviewer comment 2.5

In addition, there are two popular approaches, i.e., CCA and PLS, to implement multivariate covariance. Although there was no standard to choose which one is better for a given situation, the authors may want to simply state their considerations.

Author reply 2.5: We thank the reviewer for prompting us to clarify our considerations of choosing PLS over CCA. We note that CCA applies SVD to $(Y'Y)^{-1/2} Y'X (X'X)^{-1/2}$, so when the number of features is more than the number of samples, $Y'Y$ and/or $X'X$ become rank deficient, so matrix inversion becomes problematic⁶. However, there was no consensus on how to solve this issue with CCA. Since PLS applies SVD to $Y'X$, this is not an issue for PLS. To follow the reviewer’s suggestion here, we clarified this point in the revised supplementary materials section “**Partial least squares analysis**”:

“There are two popular approaches, i.e., canonical correlation analysis (CCA) and PLS, to implement multivariate covariance. We chose PLS over CCA because we noted that CCA applies SVD to $(Y'Y)^{-1/2} Y'X (X'X)^{-1/2}$, so when the number of features is more than the number of samples, $Y'Y$ and/or $X'X$ become rank deficient, so matrix inversion becomes problematic⁶. However, there was no consensus on how to solve this issue with CCA. Since PLS applies SVD to $Y'X$, this is not an issue for PLS.”

Reviewer comment 2.6

Furthermore, while many previous studies used the traditional PLS, some studies advocate the application of dimensionality reduction and/or regularization constraints when performing PLS. The authors opted not to perform any dimensionality reduction or to add any regularization constraint. The authors may need to make their consideration clear.

Author reply 2.6: We appreciate the reviewer's suggestion. Given the satisfactory performance of our cross-validation results (mean $r = 0.17$, permuted $p < 3.0 \times 10^{-3}$), we decided against implementing any dimensionality reduction or introducing additional regularization constraints. Our aim was to avoid the introduction of more hyperparameters such as the number of principal components or the level of sparsity that would require further tuning thus requiring potentially larger sample sizes and also leading to higher computational costs. We clarified our consideration in the revised supplementary materials section Partial least squares analysis:

“Moreover, considering that our cross-validation results were already successful (mean $r=0.17$, permuted $p < 3.0 \times 10^{-3}$), we opted not to perform any dimensionality reduction or to add any regularization constraint to avoid additional tuning of hyperparameters (e.g., number of principal components, level of sparsity, etc.) that generally requires larger sample size and have higher computational costs.”

Reviewer comment 2.7

The strength of the present study includes the utilization of a large sample, advanced

multivariate methods, and replication of results in an independent sample. A relative weaker point of the present study, in my opinion, is the relatively small sample size of the classification dataset with 52 insomnia patient and 49 healthy controls. Although the component identified with RSFC not only show macromolecular relevance, but also generalized well to unrelated individuals in the HCP dataset and were robust across alternative methodological strategies, a burning question is whether the component identified with RSFC can differentiate insomnia patients from healthy controls in another independent dataset, in other words, can the classification performance of the sleep-health related connectome generalized to another independent dataset? If the classification performance based on the sleep-health related connectome can also be generalized to another classification dataset, then we can say the results are robust and the RSFC pattern derived from the PLS analysis can be regarded as biomarker for sleep health.

Author reply 2.7: We would like to thank the reviewer for the valuable suggestion here. We have followed the reviewer's recommendation and evaluate whether the component identified with RSFC can differentiate insomnia patients from healthy controls in another independent dataset. Luckily, our collaborators (Professor Kai Yuan and Dahua Yu) shared with us their dataset from the Second Hospital of Hebei Medical University, which contained 35 insomnia patients and 35 healthy controls (referred as external classification dataset). We replicated the classification procedure conducted in the classification dataset with this external classification dataset. It turned out that the classification performance based on the sleep-health related connectome can also be generalized to this external classification dataset with an average accuracy across 100 CV of 78.29% (permuted $p < 0.0001$) with 80.00% sensitivity, 77.14% specificity and area under the curve (AUC) = 0.79 (figure S5a, green receiver operating characteristic (ROC) curve in figure S5b). We also tested to which extend the sleep-health related connectome does as well or better than using the whole brain connectome for the classification performance in the external classification dataset. It turned out the whole brain connectome-based classification performance was lower as indicated by an average accuracy across 100 CV of 74.27% (permuted $p < 0.0001$) with 74.29% sensitivity, 77.14% specificity and AUC = 0.75 (red ROC curve in figure s5b). Therefore, we followed the reviewer's comment to say the results are robust and the RSFC pattern derived from the PLS analysis can be regarded as biomarker for sleep health.

We have included this replication result with the external classification dataset in the revised manuscript (mainly in page15) and the sample details, data acquisition and image preprocessing about this external classification dataset in the revised supplementary materials, both were highlighted in blue.

Reviewer comment 2.8

I am also recommended to discuss the classification performance of the sleep-health related connectome findings (page15) in more depth by diving deeply into the research literature about insomnia classification based on a machine learning approach to separate subjects with and without insomnia symptoms using the RSFC features.

Author reply 2.8: We followed the reviewer's recommendation here and discussed the classification performance in more depth as indicated in the revised manuscript(page17):

“Critically, the RSFC spatial pattern of this dimension also has diagnostic potential to distinguish insomnia patients from sleep healthy subjects with an accuracy of 79.73% and 78.29% in dataset4 and the external classification dataset respectively. While a previous study utilized a “resting-state” fMRI support vector machine (SVM) classifier that achieved higher classification accuracy between IDs and HCs, it included 5 minutes of wakefulness, as well as data from sleep stages 1 (S1), 2 (S2), and 3 (S3) for each subject⁷. However, this approach could not exclude the possibility of shared individualized characteristics across multiple sessions from the same subject, which could inflate the classification performance. Importantly, our classification results demonstrated that the obtained brain connectome pattern also contained pathophysiological relevance for insomnia disorder. Insomnia is a common, distressing, and clinically complex symptom, which causes difficulty initiating sleep; frequent awakening; or early-morning awakening with daytime dysfunction⁴⁸. Despite its high prevalence, insomnia often goes under-diagnosed and untreated, resulting in general fatigue and decreased productivity⁹. Based on the RSFC spatial pattern-based SVM classifier derived from the obtained sleep health dimension, we will be able to aid in the diagnosis of insomnia disorder. To this end, these findings may suggest that insomnia is a condition with a multifaceted pathophysiology spanning over deficits in multiple domains of sleep health. Ultimately, the present study can serve as a foundation to identify targets that may enable sleep-based interventions for improving both brain health and clinical outcomes.”

Minor

Reviewer comment 2.9

In Fig 5., it can help the reader if the authors can point out the brain regions with high importance scores in both the positive and negative networks.

Author reply 2.9: We have followed the reviewer’s suggestion here to point out the brain regions with high importance scores in both the positive and negative network. Please refer to the revised figure5 (also indicated as below).

Fig 5. The positive and negative RSFC loadings were summed separately for each region of Brainconnectome atlas to represent the region importance scores in positive (a) and negative network (b). RSFC, resting-state functional connectivity. INS, Insular Gyrus; PrG, Precentral Gyrus; PoG, Postcentral Gyrus; Tha, Thalamus.

Reviewer comment 2.10

Considering this study is based on the cross-section design, the limitation of casual interface should be acknowledged.

Author reply 2.10: We thank the reviewer for pointing out this limitation. In the revised manuscript, we added the limitation of causal interface as highlight in blue (page 17):

“Fourth, causality between sleep health and its associated RSFC component cannot be established in this cross-sectional study. Future studies can use a longitudinal design to test whether worse sleep health leads to alterations of this sleep-health related connectome.”

Reviewer #3 (Remarks to the Author):

Reviewer comment 3.1

This study used a large and repeated sample to identify a sleep health-related brain connectivity pattern using a multivariate approach, PLS. The results showed that a composite sleep health dimension (LV1) was reflected by the variations in connectivity patterns, and this finding was replicated in two other datasets. In addition, the identified sleep-health related brain connectome showed possibility for diagnosis of insomnia disorder. This is an interesting paper that improves our understanding of the link between sleep-health and brain connectivity.

Author reply. We are glad that the reviewer found our paper interesting, and our paper improves the understanding of the link between sleep-health and brain connectivity. We are also encouraged by the reviewer’s thoughtful comments to improve the paper.

I have the following comments and suggestions for the authors:

Reviewer comment 3.2

Although the sleep-health component LV1 can interpret 28.6% of RSFC-sleep covariance, whether this component can be the best to represent for sleep-health. I suggest the authors to provide evidence, such as whether the brain connectome found by LV1 is actually had higher accuracy than by PSQI to identify insomnia disorder?

Author reply 3.2: The reviewer wondered whether the sleep-health component LV1 can be the best to represent for sleep-health. This question actually shifted our research focus, which is more about identifying the covariance patterns between multiple-sleep health domains and distributed intrinsic functional connectivity. The sleep-health component we identified with RSFC not only show macromolecular relevance, but also had diagnostic potential, generalized well to unrelated individuals in the HCP dataset and were robust across alternative methodological strategies. The reviewer suggested us to examine whether the brain connectome found by LV1 is actually had higher accuracy than by PSQI to identify insomnia disorder, which we found a bit difficult to accommodate considering the brain connectome found by LV1 is based on objective neuroimaging data to represent a composite sleep health

dimension in young adults. To be noted, we have investigated the possibility of distinguishing the insomnia patients from sleep healthy controls based on brain connectome found by LV1 to show its generalizability to insomnia disorder.

Also, reviewer 2 has a relevant recommendation for us to perform PCA to sleep health behavioral measures, then computed the Pearson's correlations between the first principal component and the behavioral saliences of LV1 identified with the PLS analysis. It turned out that the first principal component of the sleep health behavioral measures was highly correlated with the behavioral saliences of the LV1, $r=0.91$, $p=1.22 \times 10^{-14}$. This high correlation thus indicated that the first latent component was robust across analytical approaches. We added this control analysis in the revised manuscript (page 8) as:

“Seventh, the first latent component was robust across analytical approaches, which was evidenced by the high correlation between the first principal component of the sleep health behavioral measures (obtained by principal component analysis) and the behavioral saliences of LV1 ($r=0.91$, $p=1.22 \times 10^{-14}$).”

Reviewer comment 3.3

The results of the correlation with neurotransmitter densities might also require validation analysis to show its specificity. For example, the authors could provide evidence that this sleep-related connectome is related to sleep-related neurotransmitter densities, but not to emotion-related or cognitive-related neurotransmitter maps.

Author reply 3.2: We thank the reviewer for this thoughtful comment. As far as we know, there isn't a neurotransmitter that only regulates one function, such as sleep, it's all a many-to-many relationship⁵. In this study, we aimed to seek the neuromolecular relevance of the brain connectome underlying interindividual differences in sleep phenotype. The reviewer suggested us to show the specificity of the correlated neurotransmitter densities to sleep. Following this suggestion, we compared the sleep health-related connectome to the general “cognition” map that exist in Neuromaps¹⁰. This general “cognition” map has been created by taking the first principal component of NeuroSynth-derived cognitive activation¹¹. We calculated the spatial correlation between the general “cognition” map and the neurotransmitter systems that are significantly correlated with the sleep-health related connectome, including the 5HT1a, 5HT2a, GABAA and mGluR5. It turned out that the general “cognition” map was negatively correlated with 5HT1a ($r=-0.36$, $p<0.001$), and not correlated with 5HT2a ($r=0.02$, $p>0.05$), GABAA ($r=-0.02$, $p>0.05$) and mGluR5 ($r=-0.04$, $p>0.05$). Together, these results can suggest that the sleep-related neurotransmitter densities have specific correlations with this sleep-related connectome, but not to the general “cognition” (both emotion and cognition related) map.

Also, for your reference, reviewer 2 pointed out that the motivation to look into the sleep-health related neurotransmitters is not so straightforward, which may lead to such a confusion. So, we have added more information regarding why we choose to calculate the spatial correlation between the sleep-health related connectome and the distribution of several neurotransmitter systems based on the receptor maps in the revised manuscript (page 5), detailed as:

“Given accumulative evidence implicated neurotransmitter systems including serotonin receptors, glutamate and γ -aminobutyric acid (GABA) receptors were affected in sleep disturbances or “unhealthy” sleep²³⁴, we further wonder what is the neuromolecular relevance (based on neurotransmitter receptor maps⁵) of the brain connectome underlying interindividual differences in sleep phenotype?”

Taken together, we hope this can made our point clear to you.

Reviewer comment 3.4

The accuracy for distinguishing insomnia patients is less than 80%. I suggest the authors test the top ten strongest weighted degree, five positive and five negative, which is also more likely to have clinical translation potential.

Author reply 3.4: Firstly, we have followed the reviewer’s suggestion to investigate the possibility of distinguishing the insomnia patients from sleep healthy controls based on the top ten strongest weighted degree (five positive and five negative) related connectome. Specifically, significant FCs from Figure 3a associated with the top ten weighted degree nodes from Figure 3b were selected as features (698 connections out of 5956 or 11.74%). SVM achieved an accuracy of 71% > 70%, indicating that these 1/10 key features indeed possess certain discriminative capabilities. However, it is important to note that even the seemingly fewer contributory features still played a crucial role.

Secondly, reviewer 2 recommended us to test whether the component identified with RSFC can differentiate insomnia patients from healthy controls in another independent dataset. For your reference, we replicated the classification procedure conducted in the classification dataset with an external classification dataset. It turned out that the classification performance based on the sleep-health related connectome can also be generalized to this external classification dataset with an average accuracy across 100 CV of 78.29% (permuted $p < 0.0001$) with 80.00% sensitivity, 77.14% specificity and area under the curve (AUC) = 0.79 (figure S5a, green receiver operating characteristic (ROC) curve in figure S5b). We have included this replication result with the external classification dataset in the revised manuscript (mainly in page15) and the sample details, data acquisition and image preprocessing about this external classification dataset in the supplementary materials, both were highlighted in blue.

Thirdly, we have also discussed the classification performance of the sleep-health related connectome findings in more depth to follow reviewer 2’s suggestion (page 17 in the revised manuscript). Although one relevant previous study utilized a “resting-state” fMRI SVM classifier that achieved higher classification accuracy between IDs and HCs, it included 5 minutes of wakefulness, as well as data from sleep stages 1 (S1), 2 (S2), and 3 (S3) for each subject⁷. However, their approach could not exclude the possibility of shared individualized characteristics across multiple sessions from the same subject, which could inflate the classification performance.

Thus, considering the current state of the art and the evidence of robustness of our results, we think that our model achieves relatively good performance.

Reviewer comment 3.5

The discovery dataset and the replication dataset are separated by one year. Please clarify if there is a subject replication or if all participants are new.

Author reply 3.5: Actually, all participants are new. These participants were from the BBP, which was launched in September 2019 (still in progress) to recruit participants from that year's freshmen at Southwest University (SWU), Chongqing, China. Thus, the discovery dataset mainly consisted of participants from 2019's freshmen at SWU and the replication dataset mainly consisted of participants from 2020's freshman at SWU. We clarified this point in the revised manuscript (page 19) as:

“To create two independent samples for discovery and replication analyses, we split the participants according to their data collection periods (September–December 2019 [discovery dataset], namely freshman enrolled in year 2019; September–December 2020 [replication dataset] and freshman enrolled in year 2020).”

Reviewer comment 3.6

The number of permutation and bootstrap tests for the PLS analysis was 1000 and 500, respectively. However, considering the large number of participants and connections, the number of tests used here seems relatively small. It is recommended to increase the number of tests to improve the credibility of the results.

Author reply 3.6: We thank the reviewer for this thoughtful suggestion to increase the number of tests to improve the credibility of the results. As indicated in the revised supplementary materials (section Partial least squares analysis), we employed 5000 permutations (instead of 1000 permutations) and found that the LV1 remained significant, with the p-value changed from 9.9×10^{-4} to 6.0×10^{-4} . Additionally, we increased the number of bootstrap iterations to 5000 to evaluate the bootstrap-estimated standard deviation used for calculating Bootstrapped Z scores. Then, we converted the Z scores to p-values, which were FDR corrected ($q < 0.05$).

We observed that the significant patterns in both the behavior and RSFC composites remained nearly unchanged after 5000 iterations when compared to the previous 500 iterations (after bootstrap resampling and FDR correction $q < 0.05$). The significant behavioral variables remain unchanged, and 5946 / 5982 (99.40%) of the original significant RSFC connections were still significant. We have made the corresponding changes in the revised manuscript, which were highlighted in blue.

In line with this, we recalculated the cross-dataset replicability and the predictive utility of the sleep-health related connectome for both healthy population and sleep disorders. Accordingly, we have revised Figure 2b, Figure 2e, Figure 3a, Figure 7b, Figure 7c, Figure 8a, Figure 8b, Figure S1c, Figure S1e, Figure S4c, Figure S4e, Table S4 and we have added a new Figure S5 to

present the predictive utility of the sleep-health related connectome for sleep disorders with the external classification dataset.

Reviewer comment 3.7

In the PLS analysis, the z-score was first converted to a p-value to identify significant connections, and then FDR was used to correct for multiple comparisons. However, because PLS analysis is computed in a single analytical step, corrections for multiple comparisons are not required. Instead, the z-score threshold can usually be set to identify significant connections. Therefore, the analysis steps used here do not seem to reflect the advantages of PLS analysis. Please clarify the consideration of FDR correction here.

Author reply 3.7: We thank the reviewer for prompting us to clarify the consideration of FDR correction in the PLS analysis. In the PLS literature, the bootstrap is computed to "determine the elements whose responses show reliable experimental effects, thus no corrections for multiple comparisons are necessary since no statistical test is performed"¹². However, we sought to determine the z-scores' statistical significance in addition to their reliability/stability, so we computed a p-value for each z-score (i.e., a separate test for each z-score), which made it necessary to correct for multiple comparisons. We have added the clarification in the revised supplementary file.

Reviewer comment 3.8

In the external validation of the prediction, the performances of the PSQI prediction using all the connections and the connections identified in PLS were 0.16 and 0.18, respectively. If the authors want to conclude that the sleep-related connections performed BETTER, please add statistical evidence, such as the z-test between these two rs. Also, there do not seem to be any indicators similar to the MAE reported in the manuscript, so it is impossible to know how much the actual value deviates from the predicted value.

Author reply 3.8: We followed the reviewer's suggestion here to use z-test between these two rs to see whether the sleep-related connections performed BETTER than the whole brain connections¹³. It turned out the null hypothesis retained with $z=-0.3360$, $p\text{-value} = 0.3685$, which indicating there was no significant difference between the performances of the PSQI prediction using all the connections and the connections identified in PLS. In line with this, we avoided stating that the sleep-related connections performed better than the whole brain connectome in the revised manuscript. Instead, we concluded that the sleep-related connectome appears to capture relevant information for a predictive model achieving similar performance as a whole-brain connectome while using less features hence reducing the number of features to focus on for studies on smaller sample size (such as future studies in clinical samples).

We also thank the reviewer for prompting us to report the MAE in the manuscript, the MAE for the PLS-informed prediction is 2.39 and the MAE for the prediction based on the whole-brain-connections is 2.33.

Reviewer comment 3.9

Please add the range of parameters C and γ used in clinical diagnostic validation.

Author reply 3.9: We have followed the reviewer's suggestion to add the range of parameters C and γ used in clinical diagnostic validation in the revised manuscript (page 25) as:

“We used the recommended exponentially growing sequences of parameters of C and γ by LIBSVM group, i.e., C ($2^{-5}, 2^{-3}, \dots, 2^{15}$) and γ ($2^{-15}, 2^{-13}, \dots, 2^3$) based on practical guide to SVM¹⁴”.

Reviewer comment 3.10

For Table 1, the sleep characteristics should be reported, instead of family income per year.

Author reply 3.10: We thank the reviewer for prompting us to also report the sleep characteristics. We kept the family income per year in table1 given we conducted family income regression in the PLS analysis. We then reported all the sleep characteristic in a new Table2(page27, also indicated as below).

Table2. Mean and standard deviation (SD) of the sleep health measures in the discovery and replication dataset.

Domains of sleep health	Measures	Discovery dataset (N=687) Mean (SD)	Replication dataset(N=628) Mean (SD)	
Satisfaction with sleep/ Sleep quality	Not get enough sleep	2.11(0.74)	2.07(0.75)	
	Feelings from wake-up	2.35(0.82)	2.32(0.76)	
	Necessity of nap	3.09(0.89)	2.98(0.91)	
	Needed nap time	4.22(1.14)	4.19(1.15)	
	PSQI-T (otal score)	4.97(2.44)	5.33(2.58)	
	Subjective sleep quality	1.16(0.83)	1.10(0.84)	
Alertness during waking hours	Epworth sleepiness scale	8.91(3.38)	8.92(3.47)	
	Mind-wandering	13.06(3.67)	12.84(3.77)	
	Spontaneous mind wandering	19.11(4.88)	18.36(4.80)	
	Deliberate mind wandering	18.11(4.23)	17.85(4.49)	
	Attention-related cognitive errors	30.32(6.80)	30.60(7.87)	
	Frequency of daydream	33.19(9.07)	31.28(8.78)	
	Fatigue severity	38.96(8.53)	38.89(8.88)	
	Physical fatigue	4.21(2.41)	3.95(2.53)	
	Mental fatigue	2.63(1.17)	2.66(1.22)	
	Valid sleep cue RT	515.85(71.61)	521.43(83.61)	
	Invalid sleep cue RT	538.14(76.47)	553.40(90.26)	
	Valid sleep cue ACC	0.95(0.06)	0.95(0.06)	
	Invalid sleep cue ACC	0.94(0.07)	0.94(0.07)	
	Timing of sleep	Morningness-eveningness questionnaire	13.60(2.84)	13.22(2.84)
Sleep efficiency/ continuity	Sleep efficiency	0.94(0.06)	0.93(0.07)	
	Sleep latency	0.68(0.79)	0.73(0.80)	
	Wake-up times	0.30(0.69)	0.36(0.80)	
Sleep duration	Total sleep time	6.74(0.87)	6.86(0.94)	
Sleep deficiency	Insomnia Severity Index	7.60(4.25)	7.47(4.31)	
	Hyperarousal	31.90(9.50)	32.12(9.72)	
	Ford insomnia response to stress test	20.42(5.30)	20.78(5.26)	
	Sleep disturbances	0.86(0.50)	0.89(0.49)	
	Use of sleep medications	0.03(0.243)	0.06(0.35)	
	Daytime dysfunction	1.38(0.82)	1.75(0.83)	
	Sleep beliefs, attitudes, and habits	DBAS-total score	128.74(33.44)	129.75(32.84)
		Misconceptions about causes	5.73(1.72)	5.79(1.66)
Diminished control and predictability		3.42(1.43)	3.43(1.47)	
Unrealistic sleep expectations		4.67(1.63)	4.68(1.72)	
Misattribution of the		3.02(1.59)	3.13(1.68)	

consequences	
Faulty beliefs about sleep	4.00(1.26)
promoting practices	4.03(1.28)

Note. RT, reaction time; ACC, accuracy; DBAS, Dysfunctional Beliefs and Attitudes about Sleep Scale.

Reviewer comment 3.11

It is not necessary to mention nutrition at the beginning of the introduction.

Author reply 3.11: Following the reviewer's comment, we did not mention nutrition at the beginning of the introduction in the revised manuscript. So, the beginning sentence was changed into "It is increasingly recognized that sleep health (SH) is a multidimensional construct".

Reference

1. Alexander Diaz, B. *et al.* The ARSQ 2.0 reveals age and personality effects on mind-wandering experiences. *Front. Psychol.* **5**, 1–8 (2014).
2. Plante, D. T., Jensen, J. E. & Winkelman, J. W. The role of GABA in primary insomnia. *Sleep* **35**, 741–742 (2012).
3. Monti, J. M. Serotonin control of sleep-wake behavior. *Sleep Med. Rev.* **15**, 269–281 (2011).
4. Kay, D. B. & Buysse, D. J. Hyperarousal and beyond: New insights to the pathophysiology of insomnia disorder through functional neuroimaging studies. *Brain Sci.* **7**, (2017).
5. Hansen, J. Y. *et al.* Mapping neurotransmitter systems to the structural and functional organization of the human neocortex. *Nat. Neurosci.* 2021.10.28.466336 (2021) doi:10.1038/s41593-022-01186-3.
6. Malec, L. On the rank-deficient canonical correlation technique solved by analytic spectral decomposition. *J. Appl. Stat.* **49**, 819–830 (2022).
7. He, D., Ren, D., Guo, Z. & Jiang, B. Insomnia disorder diagnosed by resting-state fMRI-based SVM classifier. *Sleep Med.* **95**, 126–129 (2022).
8. Someren, E. J. W. Van. Brain mechanisms of insomnia: new perspectives on causes and consequences. *Psychol. Rev.* **101**, 995–1046 (2021).
9. Angelova, M., Karmakar, C., Zhu, Y. E., Drummond, S. P. A. & Ellis, J. Automated Method for Detecting Acute Insomnia Using Multi-Night Actigraphy Data. *IEEE Access* **8**, 74413–74422 (2020).
10. Markello, R. D. *et al.* Neuromaps: Structural and Functional Interpretation of Brain Maps. *Nat. Methods* **19**, 1472–1479 (2022).
11. Yarkoni, T., Poldrack, R. A., Nichols, T. E., Van Essen, D. C. & Wager, T. D. Large-scale automated synthesis of human functional neuroimaging data. *Nat. Methods* **8**, 665–670 (2011).
12. McIntosh, A. R. & Lobaugh, N. J. Partial least squares analysis of neuroimaging data: Applications and advances. *Neuroimage* **23**, 250–263 (2004).

13. Diedenhofen, B. & Musch, J. Cocor: A comprehensive solution for the statistical comparison of correlations. *PLoS One* **10**, 1–12 (2015).
14. Hsu, C.-W., Chang, C.-C. & Lin, C.-J. A Practical Guide to Support Vector Classification. 1–16 (2003).

Reviewer #1 (Remarks to the Author):

I appreciate the authors' efforts in addressing the concerns

The main concern of lack of control for sleep or drowsiness in the scanner however remains.

The authors have used one of the items of a questionnaire (arsq) in an attempt to control for drowsiness

The problem is that

1) this questionnaire was not developed nor validated to evaluate participants' level of alertness

2) self reported assessment sleep and sleepiness a posteriori can often be unreliable, especially in people with sleep disturbances (I.e. sleep misperception)

I therefore believe that the findings, although interesting, do not have the level of scientific robustness needed for a large-impact publication.

Reviewer #2 (Remarks to the Author):

Dear Editor,

Thank you for the invitation to provide evaluation of the authors response to reviewer #3's comments. In general, I think the authors did a great job in addressing the comments of reviewer #3. Especially that the authors have followed Reviewer (#3)'s comment to increase the number of permutation and bootstrap tests for the PLS analysis to improve the credibility of the results.

Regarding to Reviewer #1's concern regarding the measurement of alertness during resting state scans for the manuscript, I think this is an issue for resting-state fmri studies in general¹², not specific to the present study conducted by Wang and associates. I have also checked the authors' response to reviewer #1 very carefully. Actually, the authors did sufficient work to address Reviewer #1's concern, given that

1)Although the ARSQ questionnaire was developed or validated to evaluate participants' mind wandering, its "sleepiness" dimension was validated to measure participants' level of alertness in several previous studies³⁴. Therefore, I think it makes sense for the authors to extract the dimension score of "Sleepiness" from the ARSQ 2.0 and correlated it with both the RSFC and behavior composite scores obtained from the discovery dataset.

2)We can only say that self-report assessment sleep and sleepiness a posteriori can often be less reliable than objective measurements, e.g., EEG and measures of eye closure or pupil size during rsfMRI scanning. I noticed that the authors have included this point as a limitation in the revised manuscript (page 18). Nevertheless, we should notice that self-report assessment sleep and sleepiness a posteriori for such a large unique sample containing the largest sleep health domains should be considered as a strength of the present study, in my opinion.

3)I also noticed from the authors' response to reviewers' comments, participants in the discovery and replication dataset are participants from that year's freshmen at Southwest University (also refer to Table 2) rather than people with sleep disturbances (i.e., sleep misperception).

4)It should be noted that the authors have proven that the sleep-health related connectome was also independent of the circadian timing of acquisitions. Moreover, the authors have also followed the reviewer #1's suggestion to include the time (hour) of acquisition of the resting-state scanning as covariates. This additional analysis, in my opinion, can help to make sure that the sleep-health dimension not merely a reflection of drowsiness during the resting state.

Taken together, I am not quite agreeing with the conclusion given by Reviewer #1 that the findings, although interesting, do not have the level of scientific robustness needed for a large-impact publication. On the contrary, after a further evaluation of the revised manuscript together with the authors response to reviewer #1 and reviewer #3's comments, I think the method are clearer to me and appear very strong. The component the authors identified with RSFC not only show macromolecular relevance, but also had diagnostic potential, generalized well to unrelated individuals in the HCP dataset. To this end, the imitation of lacking more objective measurements such as EEG, camera, or eye-tracking during the resting-state fMRI scanning should not influence the judgement of the robustness of the research findings as well as the scientific contributions of this interesting study.

References

- 1 Chang C, Leopold DA, Schölvinck ML, Mandelkow H, Picchioni D, Liu X et al. Tracking brain arousal fluctuations with fMRI. *Proc Natl Acad Sci U S A* 2016; 113: 4518–4523.
- 2 Liu TT, Falahpour M. Vigilance Effects in Resting-State fMRI. *Front Neurosci* 2020; 14. doi:10.3389/fnins.2020.00321.
- 3 Stoffers D, Diaz BA, Chen G, Den Braber A, Van't Ent D, Boomsma DI et al. Resting-state fMRI functional connectivity is associated with sleepiness, imagery, and discontinuity of mind. *PLoS One* 2015; 10: 1–18.
- 4 Tarailis P, Koenig T, Michel CM, Griškova-Bulanova I. The Functional Aspects of Resting EEG Microstates: A Systematic Review. *Brain Topogr* 2023; : 1–37.

Reviewer #2 Attachment on the following page

I think the author has answered our concerns well.

Reviewer #1 (Remarks to the Author):

I appreciate the authors' efforts in addressing the concerns

The main concern of lack of control for sleep or drowsiness in the scanner however remains.

The authors have used one of the items of a questionnaire (arsq) in an attempt to control for drowsiness

The problem is that

1) this questionnaire was not developed nor validated to evaluate participants' level of alertness

2) self-reported assessment sleep and sleepiness a posteriori can often be unreliable, especially in people with sleep disturbances (I.e. sleep misperception)

I therefore believe that the findings, although interesting, do not have the level of scientific robustness needed for a large-impact publication.

Author reply. We thank the reviewer for evaluating our revisions. We are glad that the reviewer found our findings interesting. Regarding the main concern of the measurement of alertness during resting state scans, we have done our best to control for this general issue existing in the resting state fmri studies.

Regarding the two problems listed by reviewer #1, we have further clarifications together with the opinion of reviewer #2 as detailed below:

1) While reading the reviewer's comment and examining the recent studies of ARSQ it was apparent that the reviewer's conclusion was due to a small oversight. Also, according to reviewer #2, "although the ARSQ questionnaire was developed or validated to evaluate participants' mind wandering, its "sleepiness" dimension was validated to measure participants' level of alertness in several previous studies³⁴". Therefore, it makes sense to extract the dimension score of "Sleepiness" from the ARSQ 2.0 and correlated it with both the RSFC and behavior composite scores obtained from the discovery dataset.

2) The reviewer argues that self-reported assessment sleep and sleepiness a posteriori can often be unreliable, especially in people with sleep disturbances (I.e., sleep misperception). In agreement with reviewer #2, we can only say that self-report assessment sleep and sleepiness a posteriori can often be less reliable than objective measurements, e.g., EEG and measures of eye closure or pupil size during rsfMRI scanning. We have included this point as a limitation in the revised manuscript (page 18). In regard to the sample, we think the reviewer overlooked that participants in the discovery and replication dataset are participants from that year's freshmen at Southwest University (also refer to Table 2) rather than people with sleep disturbances (i.e., sleep misperception). Nevertheless, reviewer #2 also noticed that self-report assessment sleep and sleepiness a posteriori for such a large unique sample containing the largest sleep health domains should be considered as a strength of the present study.

- 3) According to the opinion of reviewer #2, we have proven that the sleep-health related connectome was also independent of the circadian timing of acquisitions. Moreover, we have also followed the reviewer #1's suggestion to include the time (hour) of acquisition of the resting-state scanning as covariates. This additional analysis can thus help to make sure that the sleep-health dimension not merely a reflection of drowsiness during the resting state.

Reviewer #2 (Remarks to the Author):

I think the author has answered our concerns well.

Author reply. We are excited that our revisions addressed all the concerns, and we thank the reviewer for the insightful comments again.

Reviewer #2 (Remarks to the Author):

Dear Editor,

Thank you for the invitation to provide evaluation of the authors response to reviewer #3's comments. In general, I think the authors did a great job in addressing the comments of reviewer #3. Especially that the authors have followed Reviewer (#3)'s comment to increase the number of permutation and bootstrap tests for the PLS analysis to improve the credibility of the results.

Author reply. We are glad that the reviewer thought we did a great job in addressing the comments of reviewer #3. We thank again reviewer #3 for raising helpful suggestions and for taking time to examine the PLS analysis. It prompts us to increase the number of permutation and bootstrap tests for the PLS analysis to improve the credibility of the results. We also thank reviewer #2 for taking time to evaluate our response toward reviewer #3's comments.

Regarding to Reviewer #1's concern regarding the measurement of alertness during resting state scans for the manuscript, I think this is an issue for resting-state fmri studies in general¹², not specific to the present study conducted by Wang and associates. I have also checked the authors' response to reviewer #1 very carefully. Actually, the authors did sufficient work to address Reviewer #1's concern, given that

1)Although the ARSQ questionnaire was developed or validated to evaluate participants' mind wandering, its "sleepiness" dimension was validated to measure participants' level of alertness in several previous studies³⁴. Therefore, I think it makes sense for the authors to extract the dimension score of "Sleepiness" from the ARSQ 2.0 and correlated it with both the RSFC and behavior composite scores obtained from the discovery dataset.

2)We can only say that self-report assessment sleep and sleepiness a posteriori can often be less reliable than objective measurements, e.g., EEG and measures of eye closure or

pupil size during rsfMRI scanning. I noticed that the authors have included this point as a limitation in the revised manuscript (page 18). Nevertheless, we should notice that self-report assessment sleep and sleepiness a posteriori for such a large unique sample containing the largest sleep health domains should be considered as a strength of the present study, in my opinion.

3) I also noticed from the authors' response to reviewers' comments, participants in the discovery and replication dataset are participants from that year's freshmen at Southwest University (also refer to Table 2) rather than people with sleep disturbances (i.e., sleep misperception).

4) It should be noted that the authors have proven that the sleep-health related connectome was also independent of the circadian timing of acquisitions. Moreover, the authors have also followed the reviewer #1's suggestion to include the time (hour) of acquisition of the resting-state scanning as covariates. This additional analysis, in my opinion, can help to make sure that the sleep-health dimension not merely a reflection of drowsiness during the resting state.

Taken together, I am not quite agreeing with the conclusion given by Reviewer #1 that the findings, although interesting, do not have the level of scientific robustness needed for a large-impact publication. On the contrary, after a further evaluation of the revised manuscript together with the authors response to reviewer #1 and reviewer #3's comments, I think the method are clearer to me and appear very strong. The component the authors identified with RSFC not only show macromolecular relevance, but also had diagnostic potential, generalized well to unrelated individuals in the HCP dataset. To this end, the imitation of lacking more objective measurements such as EEG, camera, or eye-tracking during the resting-state fMRI scanning should not influence the judgement of the robustness of the research findings as well as the scientific contributions of this interesting study.

References

- 1 Chang C, Leopold DA, Schölvinck ML, Mandelkow H, Picchioni D, Liu X et al. Tracking brain arousal fluctuations with fMRI. *Proc Natl Acad Sci U S A* 2016; 113: 4518–4523.
- 2 Liu TT, Falahpour M. Vigilance Effects in Resting-State fMRI. *Front Neurosci* 2020; 14. doi:10.3389/fnins.2020.00321.
- 3 Stoffers D, Diaz BA, Chen G, Den Braber A, Van't Ent D, Boomsma DI et al. Resting-state fMRI functional connectivity is associated with sleepiness, imagery, and discontinuity of mind. *PLoS One* 2015; 10: 1–18.
- 4 Tarailis P, Koenig T, Michel CM, Griškova-Bulanova I. The Functional Aspects of Resting EEG Microstates: A Systematic Review. *Brain Topogr* 2023; : 1–37.

Author reply. We thank reviewer #2 for taking time to comment on the concern of reviewer #1 regarding the measurement of alertness during resting state scans. We appreciate that reviewer #2 thought we did sufficient work to address Reviewer #1's concern. We also appreciate that reviewer

#2 thought our study interesting and the imitation of lacking more objective measurements such as EEG, camera, or eye-tracking during the resting-state fMRI scanning should not influence the judgement of the robustness of our research findings.